# Algorithmic Primitives and Compositional Geometry of Reasoning in Language Models

## Abstract

How do latent and inference time computations enable large language models (LLMs) to solve multi-step reasoning? We introduce a framework for tracing and steering algorithmic primitives that underlie model reasoning. Our approach links reasoning traces to internal activation patterns and evaluates algorithmic primitives by injecting them into residual streams and measuring their effect on reasoning steps and task performance. We consider four benchmarks: Traveling Salesperson Problem (TSP), 3SAT, AIME, and graph navigation. We operationalize primitives by clustering neural activations and labeling their matched reasoning traces. We then apply function vector methods to derive primitive vectors as reusable compositional building blocks of reasoning. Primitive vectors can be combined through addition, subtraction, and scalar operations, revealing a geometric logic in activation space. Cross-task and cross-model evaluations (Phi-4, Phi-4-Reasoning, Llama-3-8B) show both shared and task-specific primitives. Notably, comparing Phi-4 with its reasoning-finetuned variant highlights compositional generalization after finetuning: Phi-4-Reasoning exhibits more systematic use of verification and path-generation primitives. Injecting the associated primitive vectors in Phi-4-Base induces behavioral hallmarks associated with Phi-4-Reasoning. Together, these findings demonstrate that reasoning in LLMs may be supported by a compositional geometry of algorithmic primitives, that primitives transfer cross-task and cross-model, and that reasoning finetuning strengthens algorithmic generalization across domains.

## 1 Introduction

Inference time compute has remarkably improved reasoning in large language models (LLMs), and reasoning-finetuned models like Phi-4-Reasoning substantially outperform their base counterparts on reasoning tasks they were not directly trained on (Abdin et al., 2025). However, the extent to which LLMs, especially reasoning-finetuned models, can learn generalized algorithmic capacities remains poorly understood (Eberle et al., 2025). While recent advances in mechanistic interpretability (Todd et al., 2024) point in this direction, it's unclear whether LLMs acquire universal representations of algorithmic primitives (Huh et al., 2024), whether these primitives are geometrically organized in representation space, similar to brains (Fascianelli et al., 2024), and whether LLMs solve reasoning tasks by composition of generalized algorithmic primitives, and through component reuse across tasks and models (Merullo et al., 2024). This presents a unique opportunity to understand the algorithmic basis of LLM reasoning, improvements in compositional generalization after finetuning, and the impact of finetuning on chain-of-thought reasoning (Lobo et al., 2025).

This work aims to understand how fundamental algorithmic primitives are generalized and composed to enable complex reasoning capabilities in LLMs. The work addresses three fundamental questions: (1) What are the basic algorithmic primitives that language models use in specific tasks, and across reasoning domains? (2) Do these primitives compose geometrically in neural activation space? (3) How do reasoning-enhanced models differ from base models in their primitive usage and composition? We introduce a framework for a multi-level algorithmic understanding. We first extract algorithmic primitives by clustering internal representations and interpreting the corresponding reasoning traces. We then apply function vector methods to extract primitive vectors from the models' internal representations. We induce and steer algorithmic primitives by injecting primitive vectors across the layers. To identify both task-specific and universal algorithmic primitives, we

apply the approach across domains: Traveling Salesperson Problem, 3-SAT, AIME, and graph navigation. Finally, to investigate compositional generalization as a result of finetuning, we compare algorithmic performance of base models and reasoning-finetuned counterpart, like Phi-4 and Phi4-reasoning (Abdin et al., 2025). This multi-level framework allows us to understand the geometry and compositionality of algorithmic primitives in LLM reasoning.

Our contributions include: (1) the systematic identification of cross-domain algorithmic primitives in LLMs, (2) a geometric framework for understanding primitive composition through vector arithmetic, (3) novel methodology linking explicit reasoning behaviors to internal mechanisms, (4) mechanistic evidence for compositional generalization following finetuning LLMs for reasoning, and (5) evaluation of cross-task primitive transfer and generalization.

## 2  RELATED WORK

**Interpretability for Transformers.** With increasing scale and complexity of LLMs, the challenge of understanding their predictions has become an important research direction in explainable AI and interpretability research. This has specifically targeted the analysis of internal model structure and feature relationships (Geiger et al., 2022; Eberle et al., 2022; Schnake et al., 2022), as well as representations and manifolds (Kornblith et al., 2019) including concepts (Chormai et al., 2024). Mechanistic interpretability (Sharkey et al., 2025) has focused on the extraction of circuits (Olah et al., 2020; Wang et al., 2023), feature descriptions (Hernandez et al., 2022), and causally effective representations such as function vectors (Todd et al., 2024). Combined analysis of individual neurons (Gurnee & Tegmark, 2024; Templeton et al., 2024), attention scores (Voita et al., 2019; Clark et al., 2019), and task-specific attention heads (Vig & Belinkov, 2019; McDougall et al., 2024) have further deepened our understanding of internal model processing. In parallel, free-text and chain-of-thought explanations (Turpin et al., 2023; Camburu et al., 2018; Huang et al., 2023; Madsen et al., 2024) consider the model's own natural language explanation of what it is doing and why. Gradient-based feature attributions have further enabled to scale the localization and analysis of relevant features in LLMs (Ali et al., 2022; Jafari et al., 2024).

**Function Vectors and Behavioral Interventions.** LLMs' contextualized representations such as function vectors (Todd et al., 2024) and in-context task vectors (Hendel et al., 2023; Yang et al., 2025) are shown to trigger execution of associated tasks. A broader application of such steering vectors include persona vectors (Chen et al., 2025) and representations of truthfulness (Marks & Tegmark, 2024; Bürger et al., 2024). Probing has identified associations between internal representations and features of interest (Conneau et al., 2018; Hewitt & Manning, 2019). This has also been extended to the analysis of reasoning strategies, e.g., by identifying probes involved in solving grade-school math problems (Ye et al., 2025). These analyses provide some first methods to understand how complex representational geometries and manifolds drive predictions in modern models. Related work from neuroscience shows that task-related variables are encoded in neural geometries in a format that supports generalization to novel situations (Bernardi et al., 2020). Monkey neuroscience research has linked compositional generalization in neural representational geometry with behavioral performance (Fascianelli et al., 2024). These findings suggest that function vector methods can be used to study how compositional generalization may guide algorithmic steering of LLMs, providing a conceptual starting point for our work.

**Algorithmic Evaluation and Steering of Language Models.** Multi-step planning and graph navigation are standard benchmarks for structured reasoning (Fatemi et al., 2023), yet most LLMs perform poorly and fail to generalize with growing graph complexity (Momennejad et al., 2023). Although efficient algorithms exist, recent evaluations suggest that LLMs rely on policy-dependent heuristics rather than explicit search strategies (Eberle et al., 2025). Complementary works have aimed to map a high-level causal model to an internal realization by an LLMs (Geiger et al., 2024), and prompt models to generate sets of abstract hypotheses about the tasks to improve inductive reasoning performance (Wang et al., 2024). Recent analysis of chain-of-thought outputs has uncovered sentence-level relationships, offering the localization of salient reasoning steps (Bogdan et al., 2025). Existing work either targets high-level behaviors such as politeness, creativity, or honesty using relatively coarse vector representations to steer behavior (Chen et al., 2025), or focuses on very simple input-output mapping (Todd et al., 2024). Our work focuses on algorithmic steering, targeting computational primitives and their compositions, aiming at a deep understanding of the model's internal algorithmic phenotypes and applications in future algorithmic finetuning.

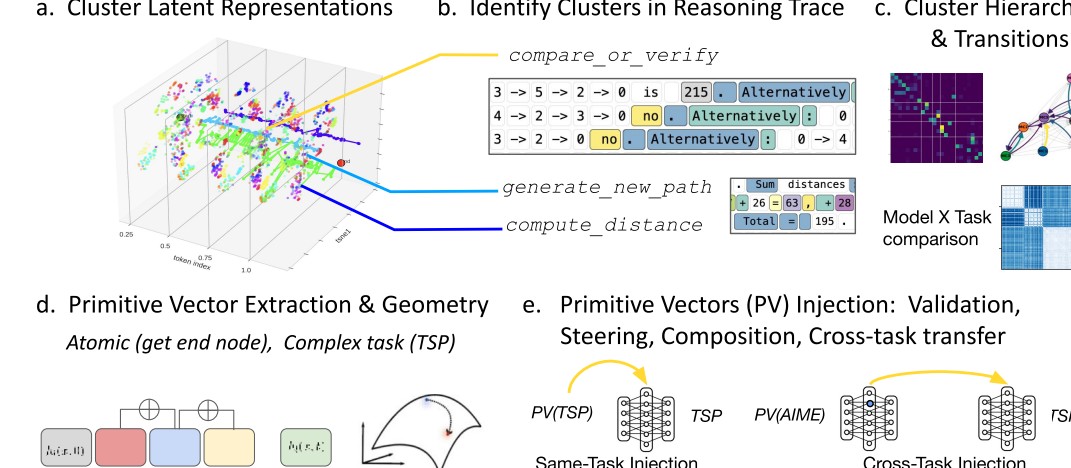

Figure 1: Algorithmic Tracing & Steering: Primitive Extraction & Evaluation. We trace algorithmic primitives by **a** clustering latent representations, **b** identifying corresponding reasoning traces, **c** meta-clustering primitives, identifying sequential transition trends among clusters, and comparing cluster similarity across models and tasks. Once we identify primitives, **d** we extract associated primitive vectors from top heads, and **e** use causal patching to validate, explore the compositional geometry and cross-task transfer of primitives.

## 3 EXPERIMENTAL SETUP: MODELS AND TASKS

**Models.** We evaluated decoder-only transformer models on multi-step reasoning tasks. Our analyses primarily focused on Phi-4-Base (Abdin et al., 2024) and the reasoning-specialized variant Phi-4-Reasoning (Abdin et al., 2025), with additional validation from Llama-3-8B (Dubey et al., 2024).

**Tasks and Data Collection.** Four multi-step reasoning setups were tested. We primarily focused on the Traveling Salesperson Problem (TSP), an NP-hard benchmark. Notably, Phi-4-Reasoning exhibited improved performance on TSP despite not being finetuned for this task (Abdin et al., 2025). We also investigated the 3-SAT and AIME Mathematical Olympiad benchmarks for the same reasons. Finally, we investigated Graph Navigation tasks used in previous LLM reasoning research (Momennejad et al., 2023; Eberle et al., 2025) (see Appendix B for task prompts). Our setups demand planning, search, and verification. TSP and Graph navigation require graph optimization via planning, path generation, search, comparison, and verification. 3SAT presents another NP-hard problem that tests flexible algorithmic problem solving. Finally, AIME requires complex mathematical reasoning and multi-step computation. Together, these setups enable us to examine algorithmic primitives that are task-specific and those that generalize across tasks. We collected reasoning traces and associated residual stream vectors across 100+ examples per task, with systematic variation in problem complexity.

## 4 METHODOLOGY

### 4.1 DEFINITIONS AND NOTATION

**Definition: Algorithmic Primitive.** We define *algorithmic primitive* as a minimal computational operation observed in a reasoning process (Eberle et al., 2025) (e.g. TSP), such as retrieving the nearest neighbor, computing a distance, generating a new candidate path, or verifying a solution. Primitives can be identified both in explicit reasoning traces (e.g. reasoning steps the model produces in the output) and in internal activations (e.g. clusters of token representations, or attention patterns).

**Definition: Algorithmic Tracing.** We define *algorithmic tracing* as the process of identifying all relevant primitives for implementing a particular reasoning process. By targeting the computational building blocks rather than just behavioral outcomes, algorithmic tracing helps us better understand how and why a model solves a particular task and what other tasks it might be able to solve.

**Definition: Primitive Vector.** We define primitive vectors (following the function and steering vector literature (Todd et al., 2024)) as vectors that can be injected into the residual stream to reliably induce a particular primitive. For a given model input $x$, which here refers to the input prompt and

Table 1: Summary of Key Terms and Definitions

| Term | Definition | Example |
|---|---|---|
| **Algorithmic Primitive** | A minimal computational operation observed in a reasoning process | Retrieving the nearest neighbor in TSP |
| **Algorithmic Tracing** | The process of identifying all relevant primitives for implementing a particular reasoning process | Solving TSP requires distance calculation, nearest-neighbor selection, path comparison, ... |
| **Primitive Vector** | A direction in activation space that can be injected into the residual stream to reliably induce a particular primitive | A vector that causes the model to use more nearest-neighbor heuristics when injected |
| **Primitive Induction** | Modifying model behavior by injecting a primitive vector | Injecting a nearest-neighbor primitive vector |

the model response, primitive $p$ and transformer layer $\ell$, let the residual stream activation at token $t$ be $h_\ell(x,t) \in \mathbb{R}^{d_\ell}$. A *primitive vector* $v_\ell^{(p)} \in \mathbb{R}^{d_\ell}$ is a direction in activation space that increases the expression of the primitive function $p$. Here we extract primitive vectors using the function vector approach (Todd et al., 2024) (see Section 4.2 for details).

**Definition: Primitive Induction for Algorithmic Steering.** Given a primitive vector injection strength $\alpha \in \mathbb{R}$, we can intervene on a representation by adding (or patching) a primitive vector:

$$\tilde{h}_\ell(x,t) = h_\ell(x,t) + \alpha\, v_\ell^{(p)},$$

which increases the probability of expressing $p$ when $\alpha > 0$, and decreases it when $\alpha < 0$. We refer to this procedure as *algorithmic steering* or *induction*.

**Algebraic Operations on Primitive Vectors.** Assuming compositional geometry, primitive vectors can be combined through simple algebraic operations in activation space. For primitives $p$ and $q$ at layer $\ell$, additive and subtractive composition within a layer (+ and –) can be defined as:

$$v_\ell^{(p \oplus q)} \approx w_p\, v_\ell^{(p)} \pm w_q\, v_\ell^{(q)}.$$

Scalar modulation (with varying strength) can be defined as:

$$v_\ell^{(\alpha p)} = \alpha\, v_\ell^{(p)}.$$

**Primitive Transfer: Cross-Task Generalization.** We test primitive generalizability by examining whether primitives identified in one domain transfer to others; specifically by examining whether injecting primitives extracted from task 1 has a predictable effect on model performance in task 2. We formalize cross-task transfer as follows. Let $p \in \mathcal{P}_{T_1}$ denote a primitive extracted from source task $T_1$, with vector $v_{\ell,T_1}^{(p)} \in \mathbb{R}^{d_\ell}$. When evaluating target task $T_2$, we inject this vector into the residual stream:

$$\tilde{h}_\ell(x_{T_2},t) = h_\ell(x_{T_2},t) + \alpha\, v_{\ell,T_1}^{(p)}.$$

If this intervention increases the activation

$$a_{\ell,T_2}^{(p)}(x_{T_2},t) = \langle v_{\ell,T_1}^{(p)},\, h_\ell(x_{T_2},t) \rangle$$

and increases the associated behavioral hallmark in task $T_2$, we denote successful transfer as:

$$p \in \mathcal{P}_{T_1} \rightsquigarrow p \in \mathcal{P}_{T_2}.$$

### 4.2 Algorithmic Tracing & Steering: Primitive Extraction & Evaluation

**Step 1. Primitive Identification: Geometric Clustering of Latent Representations.** We hypothesize that distinct algorithmic primitives are reflected in higher dissimilarities in the model's internal representations, and therefore apply k-means clustering (Lloyd, 1982) to the model's internal

representations while processing complex self-generated reasoning traces. We chose the k-means algorithm as it provided suitable results on discovering concepts in latent LLM representations in recent investigations (Hawasly et al., 2024; Petukhova et al., 2025; Kopf et al., 2025). For all clustering analyses, we extracted representations from layer 17 of Phi-4-base and used $k = 50$ clusters (we justify these choices of hyperparameters in Appendix E). We fit separate models to TSP, 3-SAT, AIME, and GraphNav and also fit a joint model to TSP+AIME.

**Step 2. Primitive Identification: Mapping Clusters to Associated Reasoning Traces.** We analyze the primitives associated with the different clusters revealed in step 1 by analyzing the tokens associated with a particular cluster and the context surrounding them. This lets us categorize algorithmic strategies, verification behaviors, and metacognitive patterns and provides direct insight into the reasoning processes models employ.

**Step 3. Primitive Composition: Hierarchical Meta-Clustering and Temporal Clustering.** To identify prevalent temporal compositions of these clusters, we then apply spectral clustering (Von Luxburg, 2007) to the matrix of transition probabilities between clusters. This gives rise to a smaller set of meta-cluster which highlight hierarchically structured reasoning steps. To infer the number of meta-clusters, we identify the largest spectral gap, using a minimum of four meta-clusters.

**Step 4. Primitive Validation: Primitive Vector Extraction.** Finally, we extract primitive vectors associated with particular clusters by adapting the methodology put forward in Todd et al. (2024). Specifically, we extract attention head activations (projected back into the residual dimension) for each response $j$, $Z^{(j)} \in \mathbb{R}^{T_j \times H \times L \times D}$ (where $T_j$ is the number of tokens of response $j$, $H$ is the number of attention heads per layer, $L$ is the number of layers, and $D$ is the residual dimension). Denoting the set of tokens that are in a particular cluster $c$ by $T_j[c] \subseteq \{1, \ldots, T_j\}$, we then compute the average attention head activations

$$\overline{Z}[c] := \frac{1}{\sum_{j=1}^{n} |T_j[c]|} \sum_{j \in T_j[c]} Z^{(j)} \in \mathbb{R}^{L \times H \times D}. \tag{1}$$

We then average these attention head activations over the top $k = 35$ attention heads that reliably carry out in-context learning functions (following the definition in Todd et al. (2024), see Appendix B.6 for further details).

This yields a candidate primitive vector for each extracted cluster. We validate candidate primitives by defining behavioral hallmarks associated with their proposed computational role and testing whether injecting the extracted primitive vectors results in an increase in those behavioral hallmarks. Beyond primitive validation, we also examine their compositional generalization by testing their arithmetic composition and cross-task transfer, injecting primitive vectors extracted from AIME into the model while it performs TSP.

**Algorithmic Fingerprinting.** For a given set of clusters $1, \ldots, k$ and a response $i$, we extract the relative frequency of tokens assigned to each cluster, $f_i \in \mathbb{R}^k$, $\sum_i f_i = 1$. We consider $f_i$ as a simple "algorithmic fingerprint" of a particular reasoning trace, highlighting differences between the primitives involved in different responses. In particular, we analyze the algorithmic dissimilarity between two responses by computing the symmetric $\chi$-squared distance between their frequencies,

$$\chi(f, g) := \sum_{i=1}^{k} \frac{(f_i - g_i)^2}{f_i + g_i}. \tag{2}$$

We also consider sets of responses $(f^{(j)})_{j=1}^{n}$, $(g^{(j)})_{j=1}^{m}$. We analyze differences in the involved algorithmic primitives by computing the average frequencies $f = \frac{1}{n} \sum_{j=1}^{n} f^{(j)}$, $g = \frac{1}{n} \sum_{j=1}^{n} g^{(j)}$ and then computing the signed $\chi$-squared distance

$$\chi_s(f_i, g_i) := \text{sgn}(f_i - g_i) \circ \frac{(f_i - g_i)^2}{f_i + g_i} \in \mathbb{R}^k, \tag{3}$$

for each cluster $i$. A large positive difference indicates that the corresponding primitive is much more prominent in the responses $(f^{(j)})_{j=1}^{m}$, a large negative difference indicates that the corresponding primitive is much more prominent in the responses $(g^{(j)})_{j=1}^{m}$. In particular, we compare Phi-4 responses to Phi-4-Reasoning responses and TSP responses to AIME responses.

## 5 RESULTS

**Primitive Tracing and Steering.** We first traced primitives by clustering latent representations over entire reasoning traces, interpreting these clusters by identifying the corresponding tokens in the model output. Perhaps surprisingly, we found that despite the simplicity of our approach, many of these clusters were highly interpretable; for example, we discovered a cluster specifically associated with the implementation of a nearest-neighbor heuristic, and another cluster associated with validations, checks, and corrections.

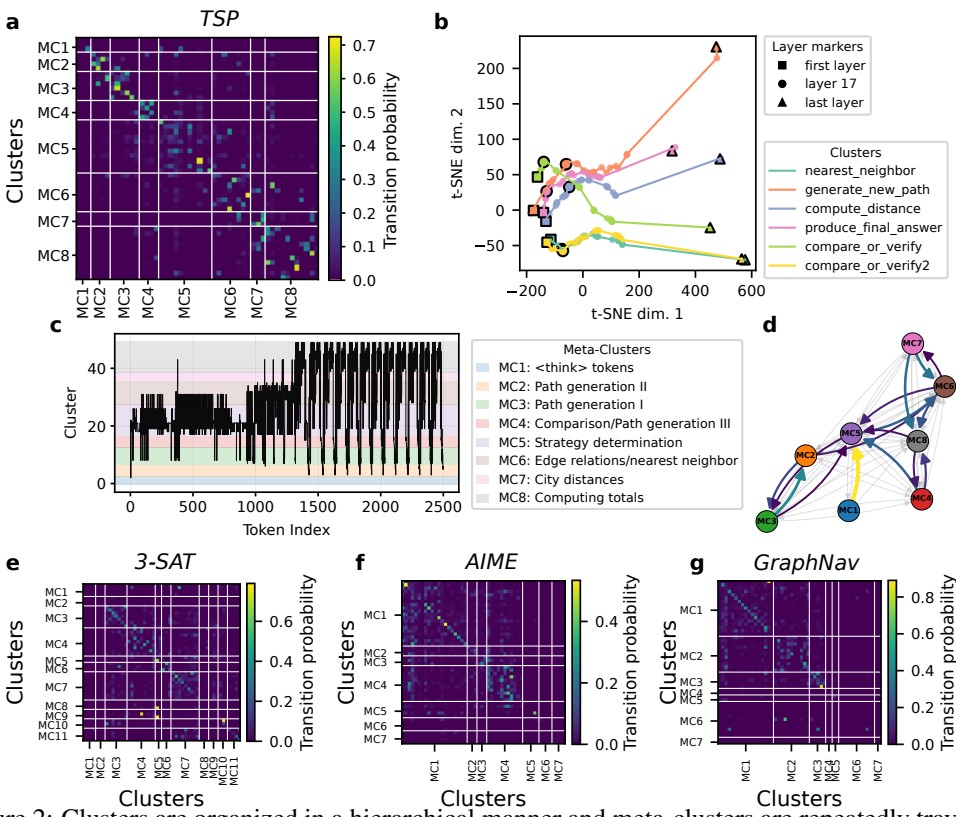

Figure 2: Clusters are organized in a hierarchical manner and meta-clusters are repeatedly traversed throughout the response. **a** Cluster-cluster transition matrix on TSP, organized by meta-cluster (MC) (indicated by the white lines). Most transitions occur within each meta-cluster. **b** Average t-SNE trajectories across different highlighted primitives. **c** Transition between different clusters for an example output. Different meta-clusters are highlighted by colors. The latter half of the response undergoes a cyclical transition. **d** Most common transitions between different meta-clusters reveals frequently occurring cycles. **e-g** Cluster-to-cluster transition matrices structured in terms of the inferred meta-clusters on **e** 3-SAT, **f** AIME, and **g** GraphNav.

We then determined the average transition probability between the different clusters. Spectral clustering of this transition matrix revealed that, on TSP, clusters are sequentially composed in a hierarchical and highly stereotyped manner, as most transitions between clusters occurred within the same meta-cluster (Fig. 2a). Notably, these different meta-clusters were also responsible for highly interpretable parts of the reasoning, e.g. path generation or computing distances (Fig. 2c).

We then validated the role of these primitives by extracting the corresponding primitive vectors and injecting them into Phi-4 during a response generation to TSP. First, we injected two vectors (`nearest_neighbor` and `generate_path`) into the model throughout the response generation, exploring different layers and magnitudes. We found that `nearest_neighbor` selectively increased the proportion of nearest-neighbor paths generated by the model, whereas `generate_path` increased the number of total generated paths (Fig. 3a,b). This demonstrates

that the extracted primitive vectors selectively induce a particular behavior and validates the candidate primitives generated by our clustering.

To expand this investigation, we injected a broader range of candidate primitive vectors into the model after providing example paths and distance computations. For example, injecting the `compute_distance` primitive causes the model to more quickly compute the distance of a candidate path. Interestingly, the reverse is also true: subtracting the `compute_distance` vector makes the model *less* likely to implement a distance computation (Fig. 3c). This suggests that subtracting primitive vectors can prevent associated algorithmic primitives. More broadly, we evaluate six behavioral hallmarks across six possible function vectors and find that all behavioral hallmarks are most strongly induced by a candidate primitive vector with a relevant computational role (Table 2).

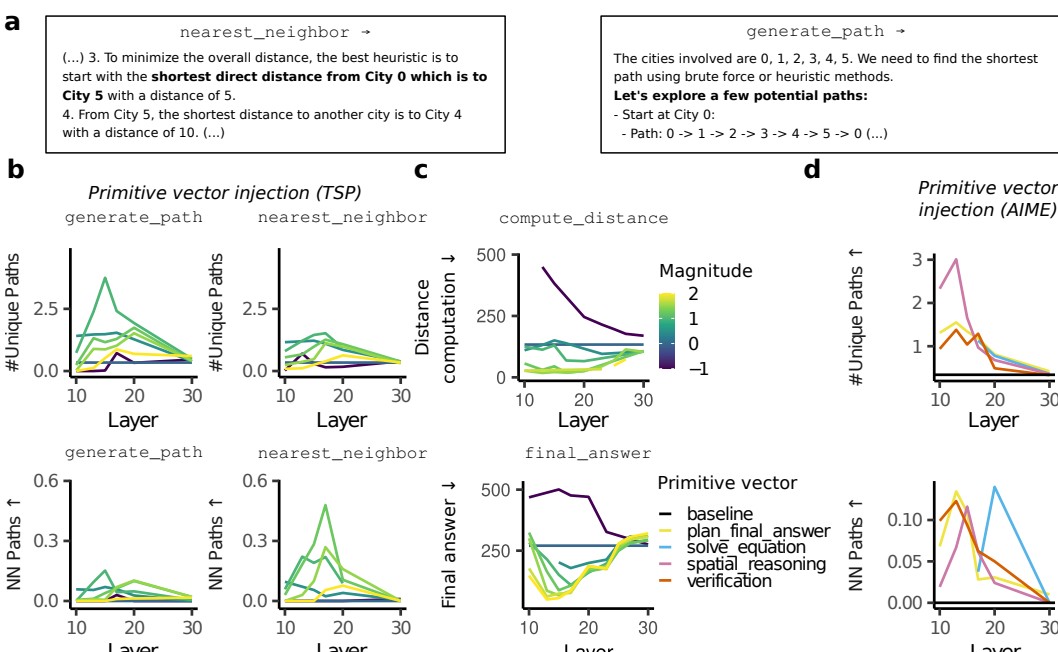

Figure 3: Primitive vector injection induces associated algorithmic behavior. Injecting the `nearest_neighbor` or `generate_path` primitive vectors directly after the prompt **a** increases expression in output (see examples). **b** Varying the injection layer and the magnitude modifies the number of unique paths generated (top row) and the proportion of nearest-neighbor paths (bottom row). **c** Injecting primitive vectors for `compute_distance` in the middle of the reasoning trace increases relevant behavioral hallmarks in the output. **d** Primitive vectors from AIME were injected to different layers while solving Traveling Salesperson (TSP), showing cross-task primitive transfer and algorithmic induction.

Notably, the role of a particular cluster cannot always be inferred from the tokens on which it is active alone; the surrounding context often also plays a role. For example, the cluster `compare_and_verify` is largely active on the token corresponding to the final distance of a candidate path. However, we noticed that tokens on which this cluster was active were often followed by subsequent checks and comparisons. We therefore hypothesized that this cluster may not just represent the final distance, but also induce a verification primitive. Table 2 confirms this hypothesis. Interestingly, we can observe this dual nature in representational space as well. In a t-SNE plot of the average representational trajectory of these example primitives, we observe that in earlier layers, `compare_or_verify` evolves along other path-related primitives like `generate_new_path`, whereas it moves closer to `compare_or_verify2` in later layers (Fig. 2b).

**Comparing Algorithmic Primitives between Phi-4 and Phi-4-Reasoning.** To identify shared and distinct primitives between responses by Phi-4 and Phi-4-Reasoning, we computed the dissimilarity between the primitive frequencies involved in different responses. We fond that responses by Phi-4-Reasoning are highly stereotyped (Fig. 4a). In contrast, Phi-4 has two subgroups of distinct responses. Notably, these groups map onto two distinct approaches towards solving TSP: a brute-force search and a random guess without further refinement. This highlights that algorithmic fin-

|  | %NN paths ↑ | #Paths ↑ | Dist. comp. ↓ | Final ans. ↓ | #Verif. ↑ | #Comp. ↑ |
|---|---|---|---|---|---|---|
| nearest_neighbor | **+56.1%** | +72.0% | -73.6% | -4.5% | +104.3% | +125.6% |
| generate_path | +4.8% | **+143.9%** | -76.0% | +18.4% | +25.0% | +12.2% |
| compute_distance | +22.2% | +36.5% | **-86.5%** | -47.4% | +198.9% | +79.3% |
| final_answer | +23.2% | +34.8% | -29.8% | **-81.5%** | +82.6% | -37.8% |
| compare_verify | +37.9% | +43.6% | -62.8% | -7.0% | +655.4% | **+1103.7%** |
| compare_verify2 | +29.5% | +48.4% | -64.9% | -13.7% | **+706.5%** | +526.8% |

Table 2: Effects of primitive vector injection on behavioral hallmarks (% above baseline). For each cell, we identify the maximal effect across all positive magnitudes and all intervention layers (10, 13, 15, 17, 20, 30). Bold = strongest effect, underline = second strongest per column. *(%NN paths: proportion of nearest neighbor paths generated. Dist. comp.: Distance computation. Final ans.: Final answer. #Verif.: number of Verifications in the output. #Comps: number of comparisons in the output. See Appendix D for detailed definitions.)*

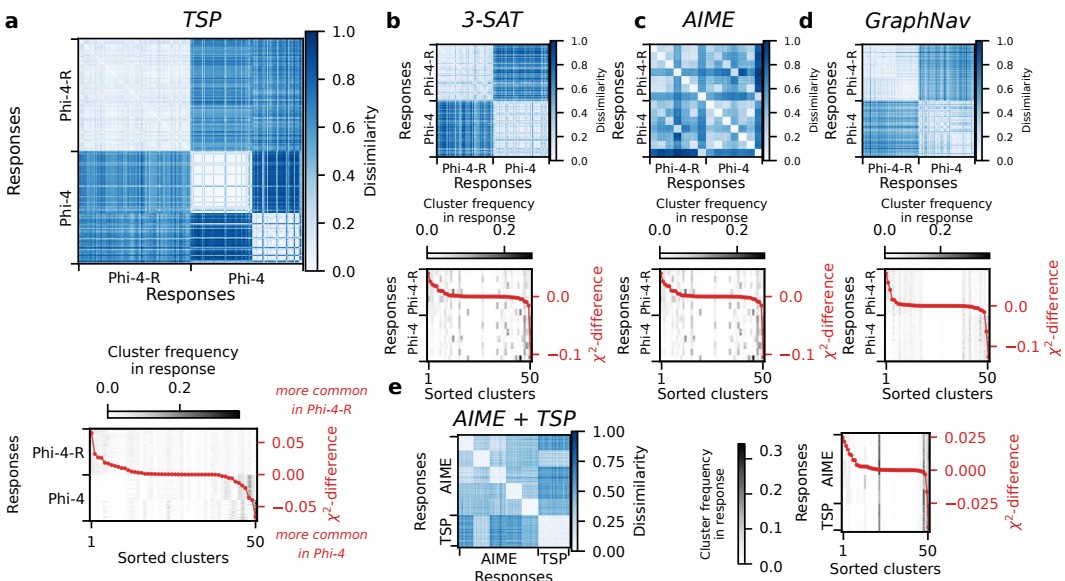

Figure 4: Primitive Cluster Patterns in Phi-4 and Phi-4-Reasoning. **a-d** *top* Normalized dissimilarity of primitive cluster frequencies between Phi-4 and Phi-4-Reasoning for a) TSP, b) 3-SAT, c) AIME, and d) GraphNav. *bottom* Clusters sorted by whether they appear more frequently in Phi-4-Reasoning responses (positive difference) or in Phi-4 responses (negative difference). The lineplot specifies the differences whereas the rasterplot underneath specifies the relative frequencies of the different clusters per response. **e** *left* Dissimilarity of primitive cluster frequencies between Phi-4-Reasoning responses to AIME and TSP. *right* Clusters sorted by whether they occur more frequently in AIME or TSP. The rasterplot underneath again specifies the relative frequencies of the different clusters per response.

gerprinting reveals both differences between different models, but even differences within responses from the same model. Beyond TSP, we found that the responses from Phi-4 and Phi-4-Reasoning on 3-SAT and GraphNav are highly stereotyped (Fig. 4b,d). On AIME, algorithmic primitives involved in responses to the same question were very similar to each other, whereas primitives involved in responses to different questions were substantially more different, reflecting the higher diversity of questions on this benchmark (Fig. 4c).

Next, we analyzed which primitives are more common in Phi-4 or Phi-4-Reasoning by computing the signed $\chi$-squared difference, and analyzing the clusters with the largest positive and negative values. This allows us to identify important differences in responses strategies across the two model. For example, on TSP and 3-SAT, distinct heuristic approaches arise more commonly in Phi-4-Reasoning than Phi-4 (nearest_neighbor and if_clause_true, Appendix A). Be-

yond task-specific strategies, our comparative analysis also highlights broader differences between the two models: in particular, clusters that are more common in Phi-4-Reasoning relate to general-purpose reasoning steps such as verification, recalling instructions, or planning the final answer.

**Compositional Primitive Induction.** So far, we have considered sequential compositions of algorithmic primitives. To investigate arithmetic compositions of primitives, we consider a set of algorithmic in-context learning tasks that require identifying 1) the last node of a path ("Terminal node recognition," TNR), 2) the node with the higher reward ("Reward comparison," RC; reward was defined by a number between 1 and 100); and 3) the most highly rewarded node between two paths, a composition of TNR and RC (see example prompt in Appendix C.1). Remarkably, we find that adding together the primitive vectors for TNR and RC induces this composite behavior in Llama-3-8B, causing it to match few-shot performance in a zero-shot setting (Fig. 5a). These effects were more attenuated in Phi-4 and Phi-4-Reasoning, highlighting that different models may require different modes of composition (Fig. 8).

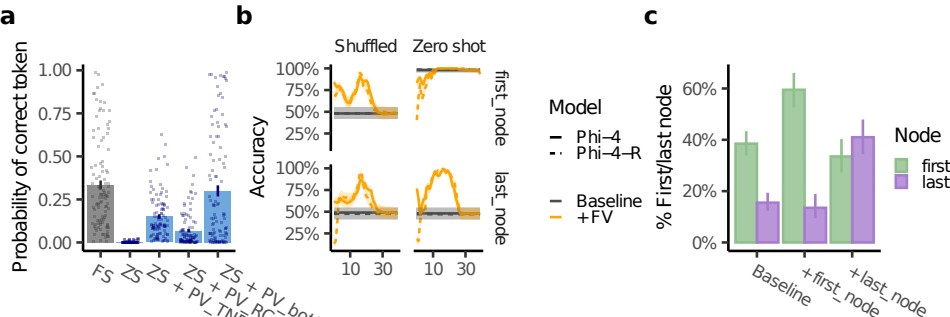

Figure 5: Algorithmic primitives operating over graphs, compositional induction, and cross-task transfer. **a** Injecting the sum of a terminal node recognition primitive vector (PV_TNR) and a reward comparison primitive vector (PV_RC) in a zero-shot setting (ZS) improves model performance on a task requiring a composition of both primitives and recovers few-shot performance (FS). See Fig. 8 for additional models. **b** Primitive vectors for extracting the first or last node of a presented graph reliably improves performance across different injection layers and across a shuffled few-shot and a zero shot-setting. **c** After injecting these primitive vectors into complex reasoning traces solving TSP, the model becomes more likely to mention the corresponding node next.

**Cross-task Transfer and Compositional Generalization.** Generalized algorithmic primitives which can be applied across different tasks, can help us understand why reasoning-finetuned models like Phi-4-Reasoning show improvements on tasks they were not finetuned on. To better understand the degree to which algorithmic primitives in Phi-4-Reasoning are shared between tasks, we apply our algorithmic tracing framework to a set of responses from both TSP and AIME (Fig. 4e). This analysis highlights that while there are different algorithmic primitives at play in either task, the two tasks also have many algorithmic primitives in common.

Next, we tested whether cross-task injection of algorithmic primitives would induce behavioral changes. First, we extracted primitive vectors from an algorithmic in-context learning task that required extracting a path's first or last node (Fig. 5b). We then injected these primitive vectors into complex reasoning traces solving TSP immediately after the model had mentioned a particular path. Despite the extremely different domain, injecting these vectors indeed had the expected effect: injecting the `first_node` (resp. `last_node`) primitive vector caused the model to subsequently mention the first node (resp. last node) of the path more often.

Finally, we considered relevant primitive vectors from AIME (e.g. `spatial_reasoning`, `plan_final_answer`) and injected them into Phi-4 generating a response to a TSP prompt. We found that these primitive vectors caused the model to generate more paths and a higher proportion of nearest-neighbor paths — a hallmark of Phi-4-Reasoning responses.

## 6 DISCUSSION

**Algorithmic Tracing and Steering.** We introduce a framework for tracing and steering algorithmic primitives as building blocks of LLM reasoning with geometric compositionality. We adapted function vector techniques (Todd et al., 2024) to extract minimal, 1-step primitive vectors from carefully designed simple tasks as well as benchmarks with more complex reasoning contexts. When "in-

jected" to LLM layers after the prompt or in the middle of reasoning traces, these primitive vectors increase the behavioral expression of corresponding reasoning steps (Figs. 3 and 5). This extends function vector research beyond simple relations between the input and output. A central contribution of our work is the compositional exploration of primitive vector arithmetic. Our framework bridges a principled path from laboratory-identified primitives to real-world reasoning behaviors.

**Toward a Geometry of Compositional Abstraction in LLMs.** We find that algorithmic primitives exhibit geometric regularities through compositional operations like addition, subtraction, multiplication, and scalar modulation. The layer and magnitude of injection shape the output expression of the primitive (Fig. 3). We show cross-task transfer and generalizability of primitive vectors: spatial reasoning and verification primitives extracted from AIME successfully transfer to TSP. This transferability hints at potentially universal algorithmic building blocks underlying diverse reasoning capabilities. If this is the case, an algorithm may be expressed by an LLM in terms of an algebraic geometry of primitive vectors and their composition.

**Reasoning vs. Memorization.** Our work contributes to recent discussions about the extent to which LLMs exhibit abstract reasoning rather than relying on memorized procedures, failing to generalize in counterfactual scenarios (Wu et al., 2024; Power et al., 2022; Zhang et al., 2024; Wang et al., 2025). While models were not directly trained on our tasks (Abdin et al., 2025), it is possible that they may be using amortized reasoning based on similar examples in the training. LLMs exhibit better problem-solving in high probability than low probability settings (e.g., McCoy et al. (2024a)), which rely on statistical regularities rather than abstract reasoning. Moreover, the absence of metacognitive abilities in LLMs results in brittle problem-solving (Johnson et al., 2024; Lewis & Mitchell, 2024). However, models finetuned for reasoning exhibit less sensitivity to task probability than base models (McCoy et al. (2024b)) and improved metacognitive-like uncertainty management strategies (e.g., verifying whether a candidate solution is correct) (Guo et al., 2025; Gandhi et al., 2025) that in turn causally improve reasoning (Bogdan et al., 2025). In line with these findings, our analysis identified algorithmic primitives for managing uncertainty (e.g., verification) and in-context recall. Such primitives and their associated behaviors may underlie a shift from memorization toward compositional problem-solving in reasoning models.

**Limitations.** One limitation is that the primitives, corresponding clusters, and meta-clusters we identify do not always map to a known algorithm. Future work can identify **primitive ontology** and algorithmic logic of different models, identifying commonalities and differences across them. Here, we focus on reasoning as multi-step problem-solving in limited tasks, which may not capture the complexity and multi-modal nature of natural reasoning, studied in cognitive sciences for decades Tversky (2005); Shepard & Metzler (1971); MacGregor & Ormerod (1996). Finally, we have focused on linear compositions of primitive vectors, leaving more complex interactions and potentially non-linear combinations of several vectors on manifolds for future work.

**Future Directions.** The algorithmic tracing and steering framework can be applied to any architecture (e.g., vision, diffusion, and multi-modal models) and domain beyond reasoning. An immediate extension is more detailed manifold analysis in order to capture the compositional geometry of primitives beyond linear composition, and establish task-specific and universal **primitive ontology**. An important future direction will be to evaluate our framework across a wider range of models. Another key direction is the **algorithmic training and finetuning of LLMs**, with algorithmic objectives. Moreover, future self-improving models can be designed for **algorithmic self-play**: generating and evaluating compositional algorithmic solutions. Finally, collecting human reasoning data on the same tasks enables us to compare and finetune model-human **algorithmic alignment**.

**Conclusion.** The identification of universal primitives and their compositional geometry opens new avenues for interpretability research and suggests principled approaches for computational models of human reasoning, model-human alignment, and enhancing LLM reasoning capabilities.

# 7 RECOMMENDED ADDITIONS COPIED FROM GUIDELINES

## 7.1 LLM USAGE DISCLOSURE

We used LLMs for generating code and for finding related work.

## 7.2 REPRODUCIBILITY STATEMENT

We provide a detailed description of our methodological implementation and will share our codebase upon acceptance.

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

## A  IDENTIFICATION OF ALGORITHMIC PRIMITIVES

### A.1  DETAILED METHODS

We implemented and analyzed five clustering models, using the following datasets:

1. five responses on TSP from Phi-4-Reasoning

2. five TSP responses from Phi-4 and five TSP responses from Phi-4-reasoning

3. five AIME responses from Phi-4 and Phi-4-Reasoning

4. five 3-SAT responses from Phi-4 and Phi-4-Reasoning

5. five GraphNav responses from Phi-4 and Phi-4-Reasoning

In all cases, we extracted the token-by-token representation from layer 17 of Phi-4-base. We used k-means clustering with $k = 50$ clusters and designed an HTML interface to visually inspect the different clusters (we will make this tool available on the project website).

Below we highlight selected clusters from the different clustering models.

### A.2  PHI-4-REASONING RESPONSES ON TSP

By manually inspecting the responses, we find that many clusters correspond to specific reasoning motifs and candidate algorithmic primitives.

**Cluster 37:** `nearest_neighbor`. This cluster is active specifically during the tokens where the model identifies the nearest-neighbor path or the closest city within a particular candidate path. As Phi-4-Reasoning's responses, in contrast to those by Phi-4, often started out by an initial guess using the nearest-neighbor heuristic (see Appendix G), we hypothesized that this would be a particularly relevant primitive.

**Cluster 26:** `generate_new_path`. This cluster usually preceded a new candidate path. We therefore hypothesized that representations in these tokens may encode relevant primitives for generating new paths.

**Cluster 15:** `compute_distance`. This cluster seems to precede the computation of the total distance of a candidate path.

**Cluster 41:** `compare_or_verify_2`. This cluster corresponds to a range of statements involved in comparisons (in particular comparing the distance of different paths), verification (e.g. verifying whether a generated path is valid), or recall (e.g. recalling the best path so far, a closely related step to comparisons).

**Cluster 35:** `compare_or_verify`. Interestingly, this cluster is usually active on the final presentation of the total distance. However, we noticed that not every token corresponding to a total distance belonged to this cluster. Rather, this cluster reliably predicted whether the next sentence involved a comparison or verification step — in particular, cluster 35 often preceded cluster 41. We therefore hypothesized that this cluster could be involved in promoting these comparisons and verifications.

**Cluster 30:** `produce_final_answer`. This cluster was active during the production of the final answer and we hypothesized that it would be relevant for that step.

### A.3  RESPONSES BY PHI-4 AND PHI-4-REASONING ON TSP

By selectively analyzing the clusters associated with the largest-magnitude differences between their involvement in Phi-4 and Phi-4-Reasoning, we identify a set of clusters that are more common in Phi-4 or Phi-4-Reasoning:

### A.3.1 CLUSTERS MORE COMMON IN PHI-4-REASONING

**Cluster 19:** `compare_or_verify`. This cluster again implements comparisons and verifications. It arises almost exclusively in Phi-4-Reasoning responses, highlighting the reasoning-finetuned model's tendency to implement more comparisons and verifications.

**Clusters 13 and 10:** `path_generation` Cluster 13 represents early cities during the path generation while cluster 10 represents the connections between cities. Notably, it only represents paths that are not generated as a part of the brute-force strategy, indicating that the computations involved in the brute-force strategy are different. For responses by Phi-4 that do not implement a brute-force strategy, these clusters are also more frequently active.

**Cluster 22:** `generate_path_guided`. This cluster precedes a path generation, but only in Phi-4-Reasoning. We therefore hypothesize that clusters 6 and 22 might induce different structures in their generated paths.

### A.3.2 CLUSTERS MORE COMMON IN PHI-4.

**Cluster 11:** `brute_force`. While this cluster sometimes arises in Phi-4-Reasoning, it is more strongly associated with Phi-4 and arises, in particular, when Phi-4 states its approach to implement a brute-force search.

**Cluster 41:** `generate_path_brute_force` This cluster is specifically active during the generation of paths involved in brute-force searches.

**Cluster 0:** `compound_distance_lookup` This cluster is active specifically before a distance lookup requires a multi-step computation. For example, consider the following segment: `**Permutation: 0 -> 1 -> 2 -> 3 -> 4 -> 5 -> 0**` – Distance = 44 `(0 to 1)` + 36 `(1 to 2)` + 32 `(2 to 3)` + 46 `(3 to 4)` + 26 `(4 to 5)` + 37 `(5 to 0)` = 221. Here, predicting the distance requires first identifying the relevant edge in the path before looking up the corresponding entry in the weight matrix. In contrast, consider the following segment: `Alternatively: 0->5->3->2->1->4->0. Then: 0->5=37, 5->3=31, 3->2=32, 2->1=36, 1->4=28, 4->0=42. Total: 37+31=68, +32=100, +36=136, +28=164, +42=206.` Here, adding the new distances only requires moving forward in the previously generated sequence and therefore does not require a multi-step computation. Importantly, cluster 0 is not active in this sentence.

### A.3.3 CLUSTERS COMMON TO BOTH MODELS.

**Cluster 27:** `edge_retrieval`. This cluster indicates the retrieval of the distance between two edges. It is a shared primitive between Phi-4 and Phi-4-Reasoning.

**Cluster 6:** `generate_path_1`. This cluster precedes a path generation and arises in both Phi-4 and Phi-4-Reasoning.

## A.4 RESPONSES BY PHI-4 AND PHI-4-REASONING ON 3SAT

**Clusters more common in Phi-4-Reasoning.** **Cluster 30:** `reasoning_scaffold`. This cluster appears to involve a lot of strategizing and reasoning.

**Cluster 37:** `if_clause_true`. This cluster corresponds to a particular strategy implemented by Phi-4-Reasoning in solving 3-SAT problems: investigating the consequences of one particular clause being true or false. Notably, this is a cluster rarely occurring in Phi-4, which may therefore illustrate an algorithmic primitive largely occurring in Phi-4-Reasoning. Indeed, we discovered these differences in strategy from our clustering analysis, illustrating how our approach could potentially support better understand differences in algorithmic strategies.

### A.4.1 CLUSTERS MORE COMMON IN PHI-4.

**Cluster 34:** `logical_expressions_latex`. This cluster is largely active on Latex based logical expressions. This is consistent with more general observations that Latex formats are much more common in Phi-4 than Phi-4-Reasoning.

### A.4.2 Clusters common to both models.

**Cluster 11:** `if_variable_true`. In contrast, this cluster mostly arises in Phi-4. Interestingly, besides being involved in general reasoning, it appears to promote an alternative strategy to the strategy above: setting particular variables to true or false. This results in a more brute-force oriented approach, mirroring the differences between Phi-4 and Phi-4-Reasoning on TSP.

**Cluster 38:** `recall_clause`. This cluster precedes the recall of particular clauses from the problem.

### A.5 Responses by Phi-4 and Phi-4-Reasoning on AIME

### A.5.1 Clusters more common in Phi-4-Reasoning

**Cluster 3:** `solve_equation`. This cluster is mostly active on two tasks which require solving an equation/inequation. While it is also partially active in Phi-4 for one of those tasks, it is much more common in Phi-4-Reasoning.

**Cluster 31:** `verification_alternate_approach`. This cluster corresponds to verifying specific statements or registering potential concerns with an approach and considering an alternative approach. It arises almost exclusively in Phi-4-Reasoning.

### A.5.2 Clusters more common in Phi-4

**Cluster 49:** `solve_equation`. This cluster is mostly active when Phi-4 solves equations, whereas it is much less active on Phi-4-Reasoning.

### A.6 Responses by Phi-4 and Phi-4-Reasoning on GraphNav

### A.6.1 Clusters more common in Phi-4-Reasoning.

**Cluster 2:** `reasoning_scaffold`. This cluster is involved in structuring the overall reasoning in Phi-4-Reasoning.

**Cluster 22:** `plan_final_answer`. Phi-4-Reasoning commonly plans its final answer during the thinking section. This cluster is active in that section.

**Cluster 9:** `recall_instructions`. This cluster is involved in Phi-4-Reasoning recalling its instructions.

### A.6.2 Clusters more common in Phi-4.

**Cluster 23:** `breadth_first_search`. This cluster is specifically active when the model plans and implements a breadth-first search algorithm. This indicates that Phi-4 is more likely to do this.

**Cluster 48:** `reasoning_scaffold`. This cluster is involved in structuring the overall reasoning in Phi-4, but not Phi-4-Reasoning.

### A.7 Cluster expressivity

To ease the automated evaluation of identified clusters, we introduce a novel metric to quantify the expressivity of each cluster. "Cluster expressivity" measures the number of unique tokens expressed by a cluster, divided by the radius of that cluster in latent space. More formally, consider a response with tokens $T \in \mathcal{T}^n$ where $\mathcal{T}$ is the set of tokens. Let $x \in \mathbb{R}^{n \times d}$ be its latent representation and $C \in \mathcal{C}^n$ be its associated cluster identities. For each $c \in \mathcal{C}$, let $\text{cent}[c] \in \mathbb{R}^d$ be the associated centroid. We then define the radius

$$\text{radius}[c] := \max_{n : C_n = c} \|x_n - \text{cent}[c]\|_2 \tag{4}$$

as the maximal $\ell_2$-distance of a data point within this cluster. We define the number of unique tokens expressed by the cluster as

$$m[c] := |\{t \in \mathcal{T} : \exists_n C_n = c, T_n = t\}|. \tag{5}$$

We then compute "cluster expressivity" as

$$\frac{\text{radius}[c]}{m[c]}. \tag{6}$$

To summarize this metric, we compute its median value across all responses. We reason that high expressivity indicates better-defined operations with rich vocabulary, i.e. multiple valid ways to express the same operation. We therefore take them as more likely to be accurate primitives with clear algorithmic meaning. Table 5 shows metrics of expressivity for all clusters extracted from the responses by Phi-4-Reasoning on TSP. Notably, our selected primitives consistantly have high cluster expressivity.

# B  EXPERIMENTAL SETUP DETAILS

## B.1  TRAVELING SALESPERSON PROBLEM

**Prompt:** The traveling salesman problem (TSP) is a classic optimization problem that aims to find the shortest possible route that visits a set of cities, with each city being visited exactly once and the route returning to the original city.

You must find the shortest path that visits all cities. The distances between each pair of cities are provided. Please list each city in the order they are visited. Provide the total distance of the trip. The final output of the result path and total distance wrapped by the final answer tag, like {<final_answer>{'Path': '0->1->2->...->N->0', 'TotalDistance': 'INT_TOTAL_DISTANCE'}</final_answer>}

The distances between cities are below: The path between City 0 and City 1 is with distance 44. The path between City 0 and City 2 is with distance 45. The path between City 0 and City 3 is with distance 45. The path between City 0 and City 4 is with distance 42. The path between City 0 and City 5 is with distance 37. The path between City 1 and City 2 is with distance 36. The path between City 1 and City 3 is with distance 27. The path between City 1 and City 4 is with distance 28. The path between City 1 and City 5 is with distance 29. The path between City 2 and City 3 is with distance 32. The path between City 2 and City 4 is with distance 38. The path between City 2 and City 5 is with distance 42. The path between City 3 and City 4 is with distance 46. The path between City 3 and City 5 is with distance 31. The path between City 4 and City 5 is with distance 26.

## B.2  AMERICAN INVITATIONAL MATHEMATICS EXAMINATION (AIME)

LLMs were presented with problems from the AIME benchmark, which tests various domains of mathematical reasoning such as algebra, geometry, number theory, and combinatorics. The correct answer for all AIME questions is an integer from 0-999.

**Example prompts (from 2025 AIME I):**

- **Problem 1:** Find the sum of all integer bases $b > 9$ for which $17_b$ is a divisor of $97_b$.

- **Problem 2:** On $\triangle ABC$ points $A$, $D$, $E$, and $B$ lie in that order on side $\overline{AB}$ with $AD = 4$, $DE = 16$, and $EB = 8$. Points $A$, $F$, $G$, and $C$ lie in that order on side $\overline{AC}$ with $AF = 13$, $FG = 52$, and $GC = 26$. Let $M$ be the reflection of $D$ through $F$, and let $N$ be the reflection of $G$ through $E$. Quadrilateral $DEGF$ has area 288. Find the area of heptagon $AFNBCEM$.

## B.3  3-LITERAL SATISFIABILITY PROBLEM (3SAT)

Models were tasked with determining the satisfiability of various 3SAT problems, a class of NP-hard problems requiring combinatorial reasoning. If a model responded that a problem was satisfiable, they were required to provide the specific literals that would achieve satisfiability.

### B.4    GRAPH NAVIGATION (GRAPHNAV)

Models were tasked with identifying the shortest path between two nodes in binary trees of varying depth (2-6, corresponding to 7-127 nodes). Each node was a randomly generated integer from 1-200, and edge lists were presented in randomized order. We included a 'forward' condition, where the initial node was the root and the goal was a randomly selected leaf, and a 'reverse' condition where the initial node was a randomly selected leaf and the goal was the root.

**Forward Direction Prompt Example:** Given the following list of connected rooms, someone wants to get to 91 from 114. The initial room and other rooms are denoted by numbers. `114->45`, `114->90`, `45->167`, `45->91`, `90->49`, `90->9`. Starting at 114, what is the shortest path of rooms to visit if someone wants to arrive at 91? Include the final response in parentheses as the list of rooms separated by commas.

**Reverse Direction Prompt Example:** Given the following list of connected rooms, someone wants to get to 63 from 119. All of the rooms are denoted by numbers. `164->63`, `119->147`, `52->147`, `54->164`, `147->63`, `62->164`. Starting at 119, what is the shortest path of rooms to visit if someone wants to arrive at 63? Include the final response in parentheses as the list of rooms separated by commas.

### B.5    GENERATED MODEL RESPONSES

For TSP, AIME, and 3SAT we used previously generated model responses (Balachandran et al., 2025); for GraphNav, we generated responses from Phi-4 and Phi-4-Reasoning ourselves, using default generation parameters (temperature: 0.8, top_k: 50, top_p: 0.95).

### B.6    PRIMITIVE VECTOR EXTRACTION

To extract primitive vectors we first compute the average indirect effect (AIE), as described in Section 2.3 of Todd et al. (2024), averaging over the same set of six in-context learning tasks (antonym, capitalize, country-capital, english-french, present-past, singular-plural). We then pick the 35 attention heads with the largest AIE, denoting them by $\mathcal{A} \subseteq \{(l,h)|l = 1, \ldots, 40, h = 1, \ldots, 40\}$, where $l$ denotes the layer of the corresponding attention head and $h$ denotes its particular index. We consider a set of $n = 200$ responses in total (100 responses from Phi-4 and 100 responses from Phi-4-Reasoning). For each response $j = 1, \ldots, n$, we extract the activities in every attention head on this response, $Z^{(j)} \in \mathbb{R}^{T_j \times L \times H \times D}$, where $L = 40$ is the number of layers, $H = 40$ is the number of attention heads, $T_j$ is the number of tokens in response $j$, and $D$ is the residual dimension ($D = 5420$). Importantly, to compute $Z^{(j)}$, we project the attention head activations back into the residual dimension using the projection weights following that attention heads (this is the same method implemented by Todd et al. (2024)). Like Todd et al. (2024), we do not apply any further scaling or normalization. For a given cluster $c$, we denote all tokens that are a part of this cluster by $T_j[c] \subseteq \{1, \ldots, T_j\}$. We then compute the average activity in each attention head over all those tokens:

$$\overline{Z}[c] := \frac{1}{\sum_{j=1}^{n} |T_j[c]|} \sum_{j \in T_j[c]} Z^{(j)} \in \mathbb{R}^{L \times H \times D}. \tag{7}$$

Finally, we compute the primitive vector $v^{(p)}[c]$ corresponding to this cluster as the average across the 35 attention heads with the largest AIE:

$$v^{(p)}[c] = \frac{1}{|\mathcal{A}|} \sum_{l,h \in \mathcal{A}} \overline{Z}_{lh} \in \mathbb{R}^{D}. \tag{8}$$

### B.7    TIME COMPLEXITY

Our time complexity is determined by two main processes:

1. **CoT Generation + Activation Collection:** $O(L^2)$, where $L$ is the sequence length (number of tokens in the reasoning trace). Complexity is quadratic because Transformer self-attention computes all token-to-token interactions. This happens once per response to generate the trace and extract internal activations. Typical responses contain between 500 and 6000 tokens; at that scale, the activation collection is the most expensive step for our method, taking between 5–10 minutes on 4 A40 GPUs (we note that multiple GPUs are required to analyze long responses).

2. **Clustering:** $O(K \times N \times I)$, with $K$ the number of clusters ($K = 50$ usually), $N$ the number of datapoints (activations collected), and $I$ the number of iterations until convergence. Scaling is linear with each factor. This is much faster than activation collection: we measured 61 seconds for 10 responses with 50 clusters (linear, ĩ min with a CPU).

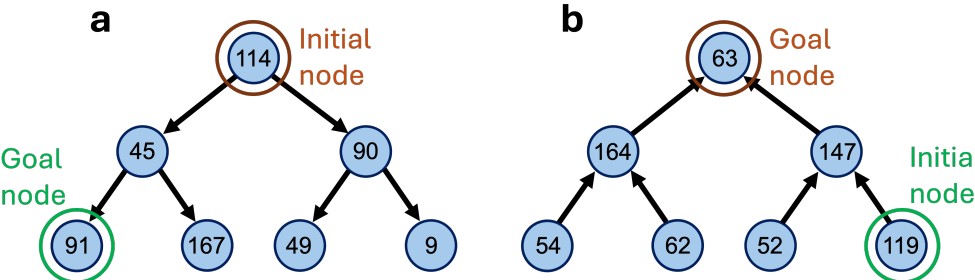

Figure 6: Examples of binary trees used in the two conditions. **a** Forward condition: models were tasked with finding the shortest path from the root node to a randomly selected leaf node, as in the forward prompt above. **b** Reverse condition: models were tasked with finding the shortest path from a randomly selected leaf node back to the root node.

## C IDENTIFYING AND COMPOSING FUNCTION VECTORS

### C.1 TASKS

We generated a set of algorithmic in-context learning tasks that require specific operations over graphs.

Terminal node recognition (TNR), identifies the final node in a path (e.g., given the input "path: T-P-Q-V", the model is tasked with outputting the token 'V'). Reward comparison (RC) compares rewards (represented by numbers between 1 and 100) across candidate nodes (e.g., given the input "rewards=[M:100 vs S:44]", the model is tasked with outputting 'M'). The evaluation task in Fig. 5a required combining both primitives: given two paths, the model was prompted to return the node that contains the highest reward.

**Example Prompt (Correct Answer: A)**

    path1: B-O-Q-D-A.
    path1-rewards=[A:65 vs Y:75].
    path2: C-W-V.
    path2-rewards [V:15 vs Y:45]

Relatedly, `get_first_node` and `get_last_node` requires responding with the first or last node of a presented path (each node consists of numbers or capital letters and has between three and six elements). `get_predecessor` and `get_successor` receive a path and a node has input and need to return the predecessor or successor node respectively.

**Example** (`get_successor`)
    *Input:* Graph: D-C-N-J, Node: C
    *Output:* N

## C.2 DETAILED METHODS AND RESULTS

We extracted function vectors corresponding to each of these tasks using the same approach as Todd et al. (2024). We identified the 35 attention heads with the highest average indirect effect (see definition in Todd et al. (2024)) for Phi-4 and Phi-4-Reasoning and 20 attention heads the highest average indirect effect for Llama-3-8B. We injected these function vectors across all layers and analyzed the layer with the largest effect.

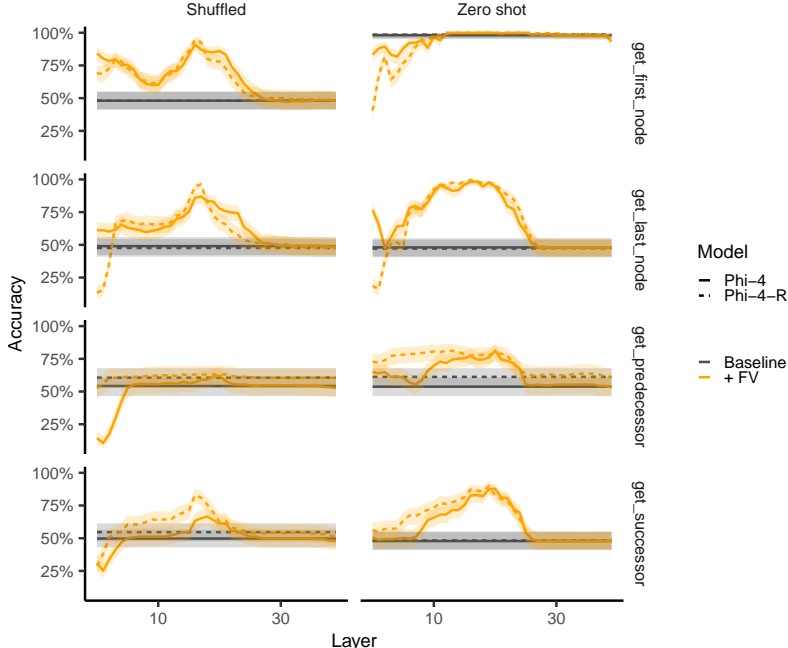

Figure 7: Accuracy on different graph operations after injecting the corresponding function vector into Phi-4 or Phi-4-Reasoning, either after shuffled in-context examples are in a zero-shot setting.

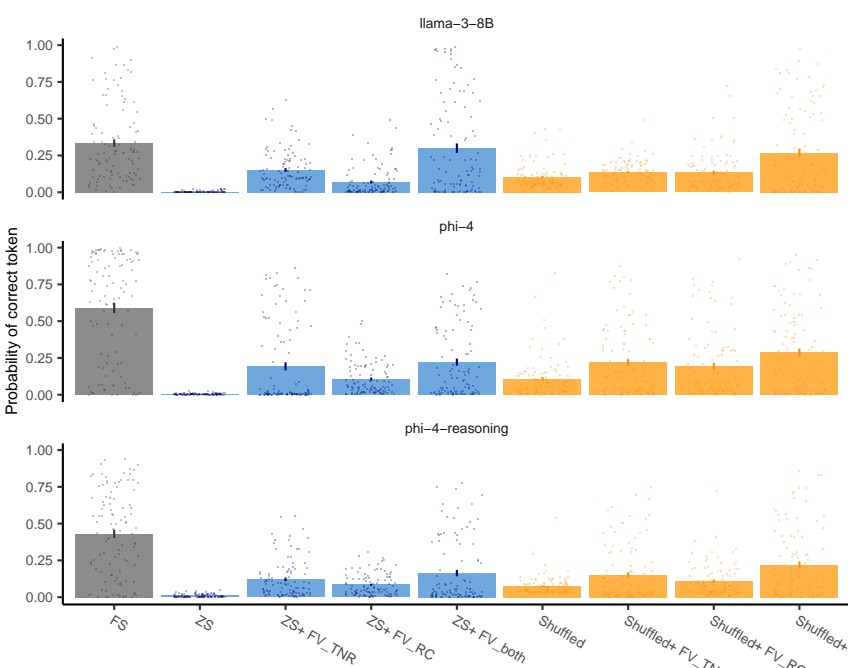

Figure 8: Average probability assigned to the correct tokens when injecting the Terminal_node_recognition and Reward_comparison FVs (FV_TNR and FV_RC) and the sum of both (FV_both). We compare the performance across Llama-3-8B, Phi-4, and Phi-4-Reasoning, and across a corrupted ICL and zero-shot context.

## D  BEHAVIORAL HALLMARKS ON TSP

We consider three setups for injecting primitive vectors into TSP. In Figs. 3b,d, we only present the model with the TSP prompt and inject the primitive vectors throughout generation. We maximally generate 500 tokens. This is not sufficient to reliably generate a full answer, but is sufficient to test if the behavior is expressed. In Fig. 3c and Table 2, we add a sequence of randomly generated paths to the assistant response before injecting the function vectors. In both of these cases, we define several measures of different behavioral hallmarks. For these measures we automatically extract the set of paths generated in the model response using a regular expression. Below we specify our operational definitions of the different behavioral hallmarks:

1. % NN paths: What proportion of generated paths corresponds to a nearest-neighbor heuristic?

2. # Unique paths: How many valid TSP candidate paths are generated (after removing duplicate paths)?

3. Distance computation: What is the earliest token at which a distance computation occurs (operationalized as a sum of several numbers)? A lower number thus corresponds to a stronger expression of this behavior. If no distance computation occurs in the entire response, we set the value to 512.

4. Final answer: At what token does the model generate the `<final_answer>` token?

5. # Verifications: How many verification-related words are mentioned in the response?

6. # Comparison: How many comparison-related words are mentioned in the response?

Finally, when evaluating the impact of the `get_first_node` and `get_last_node` primitive vectors (Fig. 5c), we automatically extract a generated path from the middle of the reasoning trace (constraining our selection to only consider paths where the first and last node are not identical). We then generate the model response starting immediately after the generated path and injecting the

corresponding primitive vector throughout. We generate 20 tokens and assess whether the first node that is mentioned corresponds to the first node of the path, the last node of the path or some other node.

## D.1 COMPARISON TO BAG-OF-TOKENS EXPLANATION

To control for the potential alternative explanation that the injected primitive vectors simply boost a "bag of tokens," increasing the probability of specific tokens, we also injected the primitive vector at the end of the residual stream, directly before the unembedding. We reasoned that if the primitive vector simply boosts a bag of tokens, the algorithmic expression should be evoked with similar strength by this procedure. We computed the maximal effect of this intervention across a wide range of magnitudes ($\alpha = 0.5, 1.0, 1.25, 1.5, 1.75, 2.0, 3.0, 4.0, 10.0, 100.0, 1000.0$) and subtracted it from the effect obtained by injecting the primitive vector into intermediate layers. We found that each primitive vector most strongly induced its corresponding behavioral hallmark even with this control, indicating that this behavior is caused by a latent procedure (Table 3).

| | %NN paths ↑ | #Paths ↑ | Dist. comp. ↓ | Final ans. ↓ | #Verif. ↑ | #Comp. ↑ |
|---|---|---|---|---|---|---|
| nearest_neighbor | **+33.2%** | +47.3% | -42.2% | -18.6% | +9.8% | +141.5% |
| generate_new_path | +0.0% | **+73.9%** | -19.7% | -0.3% | -213.0% | -7.3% |
| compute_distance | +10.1% | +33.7% | **-44.8%** | -50.5% | +128.3% | +103.7% |
| produce_final_answer | +12.8% | +34.0% | -21.2% | **-63.5%** | +47.8% | -1.2% |
| compare_or_verify | +7.9% | +38.5% | -38.0% | -20.2% | **+559.8%** | **+1059.8%** |
| compare_or_verify_2 | +5.9% | +0.8% | -24.3% | -24.7% | +516.3% | +290.2% |

Table 3: Effects of primitive vector injection on behavioral hallmarks (% above baseline), controlling for the effect of boosting a "bag of tokens". For each cell, we identify the maximal effect across all positive magnitudes and all intervention layers (10, 13, 15, 17, 20, 30) and subtract the maximal effect across all positive magnitudes of injecting the primitive vector at the end of the residual stream. Bold = strongest effect, underline = second strongest per column. *(%NN paths: proportion of nearest neighbor paths generated. Dist. comp.: Distance computation. Final ans.: Final answer. #Verif.: number of Verifications in the output. #Comps: number of comparisons in the output. See Appendix D for detailed definitions.)*

## D.2 ANALYSIS OF INTERFERENCE BETWEEN PRIMITIVE VECTORS

In Table 2 each cell had its own optimized injection magnitude $\alpha$ and injection layer $L$ (maximizing expression for that specific primitive-task combination). To investigate the specificity of algorithmic primitives, we additionally investigated the effect of injecting the algorithmic primitive at the specific $\alpha$ and $L$ that optimizes that primitive's corresponding behavioral hallmark (Table 4).

Primitive vectors were substantially more disentangled in this analysis. The analysis reveals that the entanglement of some primitives seems to follow the sequential order of their usual expression. For instance, injecting nearest_neighbor increases the expression of its algorithmic successor, compute_distance, but injecting the successor doesn't increase the expression of its predecessor (nearest_neighbor). This suggests that the entanglement of two primitives may be, at least partially, due to them being commonly expressed in specific sequential algorithmic motifs, with specific ordering.

## D.3 RECONSTRUCTING PRIMITIVE VECTORS BY THEIR TOKEN OUTPUTS

To investigate whether primitive vectors can be reconstructed by their token decodings, we follow the approach in Todd et al. (2024): for a given primitive vector $v$, we apply the LLM's unembedding matrix to generate an output distribution, $Q_t := D(v)$, where $D$ is the unembedding operation. $Q_t \in \mathbb{R}^T$ where $T$ is the number of tokens. We then truncate this distribution to the top 100 tokens and apply the softmax to generate a probability distribution $P_{t100}$ over the top-100 tokens. We then perform an optimization where $\hat{v}_{t100}$ is inferred as the vector approximating this probability distribution:

$$\hat{v}_{t100} = \arg\min_{v} \text{CE}(v, P_{t100}), \tag{9}$$

|  | %NN paths ↑ | #Paths ↑ | Dist. comp. ↓ | Final ans. ↓ | #Verif. ↑ | #Comp. ↑ |
|---|---|---|---|---|---|---|
| `nearest_neighbor` | **+56.1%** | -4.2% | -34.9% | +6.3% | +40.2% | -22.0% |
| `generate_new_path` | -41.4% | **+143.9%** | -76.0% | +85.3% | -93.5% | -82.9% |
| `compute_distance` | -100.0% | -100.0% | **-86.5%** | +37.9% | +35.9% | +79.3% |
| `produce_final_answer` | -100.0% | -95.2% | +283.8% | **-81.5%** | -48.9% | -84.1% |
| `compare_or_verify` | -100.0% | -95.5% | +266.4% | +85.5% | +568.5% | **+1103.7%** |
| `compare_or_verify_2` | -100.0% | -97.7% | +280.2% | +76.5% | **+706.5%** | +325.6% |

Table 4: Effects of primitive vector injection on behavioral hallmarks (% above baseline). For each row, we identify the magnitude and layer maximizing the effect for that row's primitive and then evaluate the effect on all behavioral hallmarks in that row. Bold = strongest effect, underline = effects that increase expression of the behavioral hallmark.

where CE is the crossentropy loss. $\hat{v}_{t100}$ thus reflects the information about the primitive vector that is contained within its decoded vocabulary. We then used those inferred vectors to generate new outputs, computing the associated effect on the different behavioral hallmarks. We found that in almost all cases, their effect was smaller than that of the original primitive vector. This indicates that knowledge of the top tokens associated with a given primitive vector are not sufficient to reconstruct its effect and some additional information is needed. A notable exception is given by the impact of `compare_or_verify` on the number of comparisons, for which the inferred vector has an even stronger effect. This suggests that the effect induced by this primitive vector is captured by its top-decoded tokens. Overall, our analysis suggests that primitive vectors usually contain some information beyond what can be inferred from their top decoded tokens.

| Metric | PV | $v$ | $\hat{v}_{t100}$ |
|---|---|---|---|
| % NN paths ↑ | `nearest_neighbor` | **+56.1%** | +21.1% |
| # Paths ↑ | `generate_new_path` | **+143.9%** | +62.0% |
| Dist. comp. ↓ | `compute_distance` | **-86.5%** | -71.9% |
| Final ans. ↓ | `produce_final_answer` | **-81.5%** | -16.2% |
| # Verif. ↑ | `compare_or_verify_2` | **+706.5%** | +71.7% |
| # Comp. ↑ | `compare_or_verify` | +1103.7% | **+2458.5%** |

Table: Performance of the primitive vector (PV) (column $v$) compared to the reconstruction $\hat{v}_{t100}$ that matches the top 100 tokens (both columns show % above baseline). For each primitive vector, we considered the behavioral hallmark on which it had the strongest effect (see Table 2). In all cases except for the number of comparisons, the original PV induces a stronger effect.

# E   HYPERPARAMETER COMPARISONS

To evaluate robustness to our chosen hyperparameters, we repeated the clustering analysis for different choices of layer, number of clusters, and number of responses.

## E.1   NUMBER OF LAYERS

We first repeated the clustering analysis for all layers by increments of five in both Phi-4 and Phi-4-Reasoning when solving Traveling Salesperson (TSP). We then compared the frequency of each layer's clusters showing up both across responses within a model, and across the responses of the two models (the analysis presented in Fig. 4a) (see Fig. 9a). To evaluate uniquely informative patterns across the layers, we computed the correlation across the layer-wise dissimilarity matrices (Fig. 9b). Together, these analyses revealed broad patterns in the effect of layers: after the first layers, there was a large highly correlated cohort in the intermediate layers, especially layers 10-25 (centered on layer 17), and a smaller cohort of highly correlated layers, ranging from 30-40. We observed that intermediate layers captured more aspects of the reasoning hierarchy, as they more strongly reflected the differences between the brute-force and random guess strategies implemented by different responses from Phi-4 (see Appendix G). In contrast, later layers more strongly highlight the

differences between Phi-4 and Phi-4-Reasoning. As we are interested in identifying the underlying algorithmic primitives, these findings justify layer 17 as a suitable candidate for clustering.

### E.2 NUMBER OF CLUSTERS

To evaluate robustness to our choice of $K$, we conducted a similar analysis, this time keeping the layer fixed (17) and varying $K$ by 5, 30, 50, and 100. As described above, we first computed dissimilarity matrices for frequency of cluster occurrences in within-model and cross-model responses per K (Fig. 10a), and then computed the correlation across these matrices (Fig. 10b). Applied to 100 responses to TSP per model (Phi-4 and Phi-4R), this analysis revealed high correlations among K=30, 50, and 100, with the highest Pearson correlation of $>0.98$ between K=50 and 100.

Next, we computed normalized inertia, or the sum of squared distance between each data point and its assigned cluster centroid for 4 tasks per $K = 5, 10, ..., 100$, with increments of 5 on the x axis (Fig. 10c). We then used the elbow method (Han et al., 2012) to identify an approximate $K$ at which inertia dropped for each task. Together, these findings revealed $K = 50$ as the parsimonious and functionally appropriate choice, offering a reasonable tradeoff between explaining variance in the latent space and parsimony in the potential number of algorithmic primitives.

### E.3 NUMBER OF SAMPLES

To evaluate robustness to sampling choice, we fixed the layer ($L = 17$) and $K = 50$, and evaluated the robustness of clustering to different numbers of responses, between 1 and 50. First, we found very high correlation between assigned clusters across varying number of responses with a minimum Pearson's correlation of 0.96 for using 1 response, and a correlation of 0.9945 between 5 and 25 or 50 responses (Fig. 11a,b). We then measured the consistency of assigned clusters across a varying number of responses with 2 measures: adjusted mutual information (MI) score and adjusted Rand score (Fig. 11c), revealing high consistency for clustering models fit to at least 5 responses. Taken together these findings show robustness of assigned clusters to varying number of responses included in the analysis.

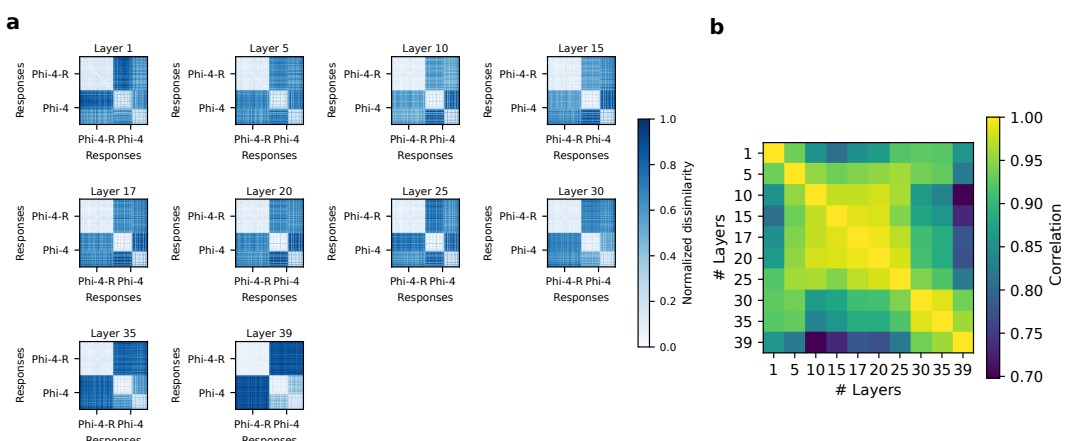

Figure 9: **a** Dissimilarity matrices between the frequency of clusters occurring in different responses, plotted for cluster analyses fitted to different layers. **b** Similarity between these different dissimilarity matrices. **c** Average dissimilarity between responses from different groups: 1) responses by Phi-4-Reasoning, which consistently employ a heuristic nearest-neighbor strategy (see Appendix G), 2) responses by Phi-4 that employ a brute-force strategy, 3) responses by Phi-4 that articulate a random guess. Intermediate layers emphasize similarities between the heuristic strategy employed by Phi-4-Reasoning and both strategies employed by Phi-4, whereas later layers mostly reflect dissimilarities between different models.

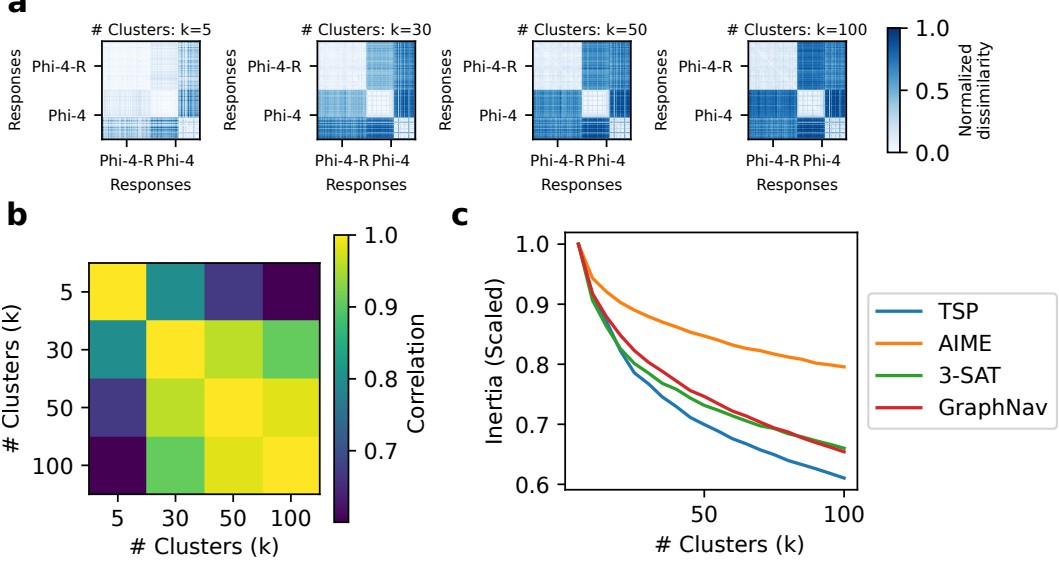

Figure 10: **a** Dissimilarity matrices as in Fig. 4a for different numbers of clusters. **b** Similarity between the dissimilarity matrices for different numbers of clusters, confirming a high degree of similarity for sufficiently many clusters. **c** Inertia (normalized by the inertia for five clusters for each task) plotted against the number of clusters across the five tasks.

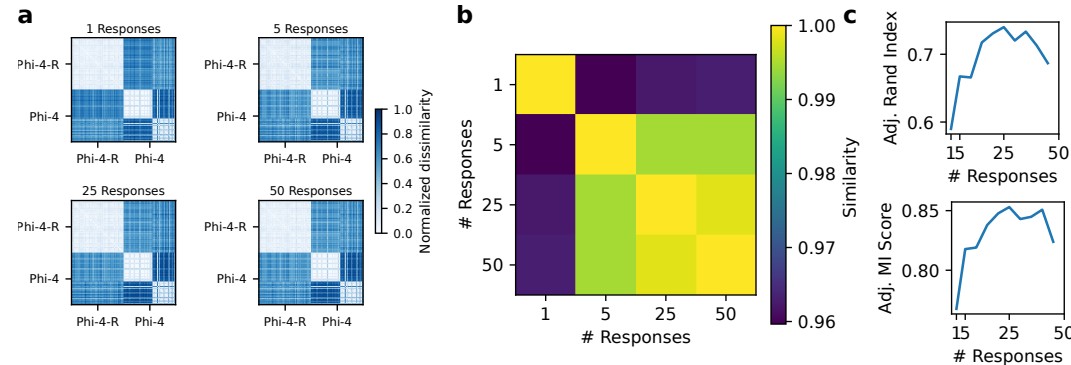

Figure 11: **a** Dissimilarity matrices between the algorithmic fingerprints of different responses, plotted for cluster analyses fitted to different numbers of responses. **b** Similarity between these different dissimilarity matrices. **c** Adjusted Rand Index and Adjusted Mutual Information Score (averaged across 100 responses) for clustering models fit to different numbers of responses.

## F   AUTOMATED LLM ANNOTATION

We developed an automated pipeline for labeling our clusters, which corroborates our manual annotations. Specifically, for each cluster we randomly sampled 50 tokens, included the 15 preceding and following tokens as context, and formatted these into 50 examples for GPT-4o. Within each example, we highlighted the critical token using the notation `**[critical token]**`. Examples were ordered by response and then by token index within each response.

We include four examples below from Phi4-Reasoning's Cluster 41, which illustrate a heterogeneous set of tokens that are all used during the evaluation and comparison of reasoning steps.

Example 1: `0->5->4->1->3->2->0 is**[ best]**.`

`Perhaps:  0->5->1->3->4->2`

Example 2: `=121, 121+30=151, so that's 151**[,]** not better.`

`What if 0->2->5->1->4`

Example 3: `90.  Remaining:  F, then F->A: 50.  But**[ wait]**, we still have F: from D, F: D->F =`

Example 4: `+12+11+45+17+21=120, we already**[ computed]** that.`

`Wait, check:  0->4->2->5->`

We employed three prompting methods to extract labels for each cluster: Prompt 1 emphasized the contextual usage of critical tokens, while Prompt 2 included each 31-token excerpt without highlighting the critical tokens. Prompt 3 served as an alternative to Prompt 1, placing greater emphasis on the critical tokens themselves, rather than their surrounding context. We found that Prompts 1 and 2 yielded the most accurate labels, aligning closely with our manual annotations. The full prompts are provided below:

Prompt 1 - Token-Centric: This text includes excerpts from a language model solving various traveling salesperson problems. It includes tokens highlighted with double asterisks and brackets (e.g., **[token]**), with the surrounding tokens provided as context. Consider the context in which each highlighted token appears, and provide a single phrase (10 words or fewer) that describes their algorithmic role or roles across the whole set. The descriptive label should be specific, because it will later be contrasted with labels for tokens from other excerpts. The label may relate to a specific type of operation, reasoning strategy, heuristic, organizational principle, class of words, numbers relevant to the problem, tokens that appear in a specific context, etc. Remember that the label should be based on the context of the highlighted tokens.

Prompt 2- Context-Centric: This text includes excerpts from a language model solving various traveling salesperson problems. Consider all of the provided examples, and provide a single phrase

(10 words or fewer) that describes the algorithmic role or roles common across the examples. The descriptive label should be specific, because it will later be contrasted with labels for other excerpts. The label may relate to a specific type of operation, reasoning strategy, heuristic, organizational principle, class of words, numbers relevant to the problem, tokens that appear in a specific context, etc.

Prompt 3 - Token-Centric (Alternative to Prompt 1): This text includes excerpts from a language model solving various traveling salesperson problems. It includes various critical tokens highlighted with double asterisks and brackets (e.g., **[critical token]**), with the surrounding tokens provided as context. Consider each highlighted token in the text, and provide a single phrase (10 words or fewer) that describes the entire set of critical tokens or encapsulates their algorithmic role or roles. The descriptive label should be specific, because it will later be contrasted with labels for tokens from other excerpts. The label may relate to a specific type of operation, reasoning strategy, heuristic, organizational principle, class of words, numbers relevant to the problem, tokens that appear in a specific context, etc. Remember that the label should be based on the highlighted tokens.

### F.1 LABELS OF ALL CLUSTERS FOR PHI-4-REASONING RESPONSES ON TSP

We used our automated pipeline to assign labels to all cluster on the clustering model fit to Phi-4-Reasoning responses on TSP (see Table 5), sorting these labels by their associated expressivity. We excluded all clusters that were expressed in fewer than two tokens on average, resulting in a total of 46 clusters. Our results highlight that our pipeline demonstrates a strong agreement with our manual labels and, furthermore, that our selected clusters generally have high associated expressivity. To better understand how many clusters correspond to distinct algorithmic primitives, we used an LLM-based pipeline to further sort them into specific groups. This revealed eight clusters that were better characterized by linguistic patterns than algorithmically distinct roles:

- **Algorithmic Clusters:**
    - High-Level Strategies: 18, 37, 41, 35, 26, 6
    - Arithmetic Operations: 15, 11, 34, 42, 4, 13, 16, 8
    - Distance/Cost Calculation: 9, 0, 17, 19, 22, 23
    - Path Construction/Evaluation: 3, 14, 1, 38, 25
    - Edge/Path Operations: 5, 27, 33, 45, 47
    - Route Optimization: 31, 39, 46, 44, 21, 10
    - Organizational/Workflow: 30, 43
- **Linguistic Clusters:**
    - Notation/Punctuation: 12, 29, 24
    - Identifiers/Labels: 7, 40, 20
    - Structural Markers: 32, 48

Table 5: Annotated labels for the clustering model fit to TSP, showing both manual labels and the two distinct automated labels. Clusters are ranked according their expressivity (number of unique tokens expressing the cluster, divided by the radius of the cluster in the latent space)

| Rank | Cluster | Manual Label | Automated label (Token-centric) | Automated label (Context-centric) | Expressivity |
|------|---------|--------------|----------------------------------|-------------------------------------|--------------|
| 1 | 18 | | Exploration of alternative routes and possibilities | Evaluating candidate routes and calculating total distances. | 2.58 |
| 2 | 30 | produce_final_answer | Output formatting with `<final_answer>` tag | Output formatting with `<final_answer>` tag. | 1.17 |

Continued on next page

Table 5: (continued)

| Rank | Cluster | Manual Label | Automated label (Token-centric) | Automated label (Context-centric) | Expressivity |
|---|---|---|---|---|---|
| 3 | 41 | compare_or_verify_2 | Route exploration and modification suggestions | Route evaluation and improvement through swaps and recalculations. | 1.11 |
| 4 | 37 | nearest_neighbor | "Selecting best path based on edge weights" | "Selecting best path based on edge costs." | 0.94 |
| 5 | 19 | | Path and distance notation | Calculating and comparing path distances. | 0.82 |
| 6 | 7 | | Key terms in TSP problem description | Shortest path computation in Traveling Salesman Problem (TSP). | 0.65 |
| 7 | 9 | | City and path identifiers | Distance calculations between city pairs. | 0.56 |
| 8 | 35 | compare_or_verify | Proposed alternative routes | Route optimization and evaluation in TSP solutions. | 0.35 |
| 9 | 15 | compute_distance | Summation and accumulation of path costs | Addition and summation of route distances. | 0.32 |
| 10 | 40 | | City and path descriptors | City-to-city distance representation. | 0.32 |
| 11 | 11 | | Intermediate cumulative sum calculations | Incremental distance calculations with comparisons to find optimal path. | 0.30 |
| 12 | 25 | | Delimiters and separators in route descriptions | Evaluating and selecting paths based on distance or cost. | 0.30 |
| 13 | 0 | | Distance values between cities | City-to-city distance calculations | 0.29 |
| 14 | 6 | | Exploring alternative routes and strategies for optimization. | Route exploration and optimization through swaps and recalculations. | 0.28 |
| 15 | 29 | | Punctuation or spacing in route calculations | Calculating and updating total distances in routes. | 0.27 |
| 16 | 17 | | Distances between city pairs | Pairwise city distances in traveling salesperson problems. | 0.25 |
| 17 | 26 | generate_new_path | Proposed routes for alternative solutions | Route exploration and evaluation for optimization. | 0.21 |
| 18 | 42 | | Intermediate sum in cumulative addition calculations | Incremental sum calculations for path cost evaluation. | 0.20 |
| 19 | 12 | | Path and distance notation in TSP solutions | Path calculation and distance summation. | 0.18 |
| 20 | 34 | | Intermediate sum values in calculations | Incremental addition of distances to calculate total path length. | 0.15 |

Table 5: (continued)

| Rank | Cluster | Manual Label | Automated label (Token-centric) | Automated label (Context-centric) | Expressivity |
|---|---|---|---|---|---|
| 21 | 45 | | Edge weight in TSP route calculations | Edge traversal and cost calculation in paths. | 0.14 |
| 22 | 24 | | Colon and equals sign for distance assignment | Node-to-node distance calculations | 0.13 |
| 23 | 22 | | City identifiers in path descriptions | Pairwise city distance calculations | 0.13 |
| 24 | 33 | | Edge weights in path calculations | Incremental cost calculations and route suggestions | 0.13 |
| 25 | 1 | | Node in a potential TSP route sequence. | Proposing alternative routes and computing distances. | 0.12 |
| 26 | 32 | | Next node in path sequence | Node-to-node distance calculations | 0.12 |
| 27 | 43 | | Instructional and structural language for problem-solving process | Structured reasoning and solution presentation. | 0.12 |
| 28 | 20 | | City identifiers in routes and calculations | Path selection and distance calculation. | 0.12 |
| 29 | 38 | | Node in a potential route path. | Evaluating and comparing potential routes and distances. | 0.10 |
| 30 | 47 | | Separating route segments and calculations | Edge weights and path calculations | 0.10 |
| 31 | 3 | | Route segment in candidate solutions | Evaluating and updating candidate routes based on distance calculations. | 0.09 |
| 32 | 5 | | Edge weights in path calculations | Edge weights and path sequences in TSP solutions | 0.09 |
| 33 | 14 | | Path traversal representation | Path construction and evaluation | 0.09 |
| 34 | 23 | | Route termination or cycle completion indicator | Proposing alternative routes and calculating their distances. | 0.08 |
| 35 | 4 | | Addition operation in arithmetic calculations | Incremental sum calculation with intermediate results | 0.07 |
| 36 | 13 | | Addition operation in distance calculations | Incremental distance calculation for route optimization. | 0.07 |
| 37 | 27 | | Edge weights in path calculations | Sequential path calculations with cumulative distance summation. | 0.07 |
| 38 | 16 | | Addition operation in cost calculations | Partial sum calculations for route cost evaluation. | 0.06 |

Table 5: (continued)

| Rank | Cluster | Manual Label | Automated label (Token-centric) | Automated label (Context-centric) | Expressivity |
|------|---------|--------------|--------------------------------|-----------------------------------|--------------|
| 39 | 31 | | Route transition indicator in path sequences | Route exploration and evaluation in TSP solutions. | 0.05 |
| 40 | 48 | | Indicating direction or transition between nodes. | Node-to-node path with distances | 0.05 |
| 41 | 8 | | Equality check in arithmetic operations | Incremental distance calculations for route optimization. | 0.05 |
| 42 | 46 | | Route transition indicator | Route exploration and evaluation for optimization. | 0.03 |
| 43 | 39 | | Route transition indicator | Route exploration and evaluation for optimality. | 0.03 |
| 44 | 44 | | Intermediate node in proposed paths | Evaluating and suggesting alternative routes for optimization. | 0.03 |
| 45 | 21 | | Return to starting point in route. | Evaluating and calculating potential routes and their distances. | 0.02 |
| 46 | 10 | | Starting point or reference city in TSP routes. | Evaluating and comparing potential TSP routes and distances. | 0.02 |

## F.2 ROBUSTNESS OF OUR PIPELINE TO DIFFERENT NUMBERS OF CLUSTERS

To explore the robustness of our pipeline to varying the number of clusters, we identified automated labels for the clusters inferred on responses by Phi-4 and Phi-4-Reasoning at $k = 100$. We analyzed the five responses that were most strongly used in Phi-4-Reasoning and Phi-4 respectively, as indicated by their $\chi^2$-difference. Notably, the top cluster in Phi-4-Reasoning expressed a nearest-neighbor approach, whereas the top cluster in Phi-4 expressed a brute-force approach. This replicates our insights at $k = 50$ and demonstrates the utility of our automated approach.

| Cluster | Automated label (Token-centric) | Automated label (Context-centric) | $\chi^2$-difference |
|---|---|---|---|
| 89 | Punctuation and separators in problem-solving context | Selecting shortest path or edge based on distance. | 0.04 |
| 67 | Exploration of alternative routes and solutions. | Route optimization through permutation and manual inspection. | 0.03 |
| 54 | Route and distance calculation punctuation | Route evaluation and distance calculation. | 0.02 |
| 22 | Comparative or superlative evaluation of routes or distances. | Route exploration and optimization in TSP solutions. | 0.02 |
| 48 | Route evaluation and comparison indicators | Route exploration and evaluation for optimization. | 0.02 |
| 1 | Placeholders for missing or irrelevant information | City-to-city distance calculations | -0.02 |
| 58 | Route representation in traveling salesperson problem solutions. | Node sequences representing potential solutions | -0.02 |
| 64 | Punctuation and formatting for path or route descriptions. | Node sequence representation | -0.03 |
| 15 | Placeholder for missing distance values | Summing distances for paths or routes. | -0.05 |
| 59 | Brute-force approach for small number of cities (6). | Brute-force permutation evaluation for small-scale TSP solutions. | -0.06 |

Table 6: Labels for clustering model fit to responses from Phi-4 and Phi-4-Reasoning with $k = 100$. We provide the five clusters with the largest positive $\chi$-squared difference (indicating that they are more often used in Phi-4-Reasoning) and the five clusters with the largest negative $\chi$-squared difference (indicating that they are more often used in Phi-4).

## G   NEAREST NEIGHBOR STRATEGY ANALYSIS

We compared the implementation of the nearest neighbor search heuristic on the Traveling Salesperson Problem (TSP) in Phi-4-Reasoning and Phi-4-Base. First, we computed the minimum edit distance between nearest neighbor solution and the optimal solution and analyzed how this distance relates to accuracy. We computed these minimal NN-optimal edit distances using any city as the starting point (Figure 12 a) and with city 0 as the starting point (Figure 12 b). The latter edit distance was chosen, because in practice, we observed that the models frequently used city 0 as the initial node.

Next, we tested whether one of the first five generated paths is a nearest-neighbor solution; we sampled from the first five paths to account for idiosyncrasies in the model's output (e.g., repeating path segments from the prompt). Additionally, we examined the relationship between the number of considered paths and the average distance from the NN solution.

Overall, these findings demonstrate that Phi-4-Reasoning has a strong tendency to implement the nearest neighbor search heuristic. Specifically, the model tends to begin with the NN solution starting at city 0 and iteratively edit the path to minimize its total distance. While Phi-4-Base does not invoke the nearest neighbor heuristic as reliably as Phi-4-Reasoning, it does to some degree as shown in Figures 12 b and 13 a.

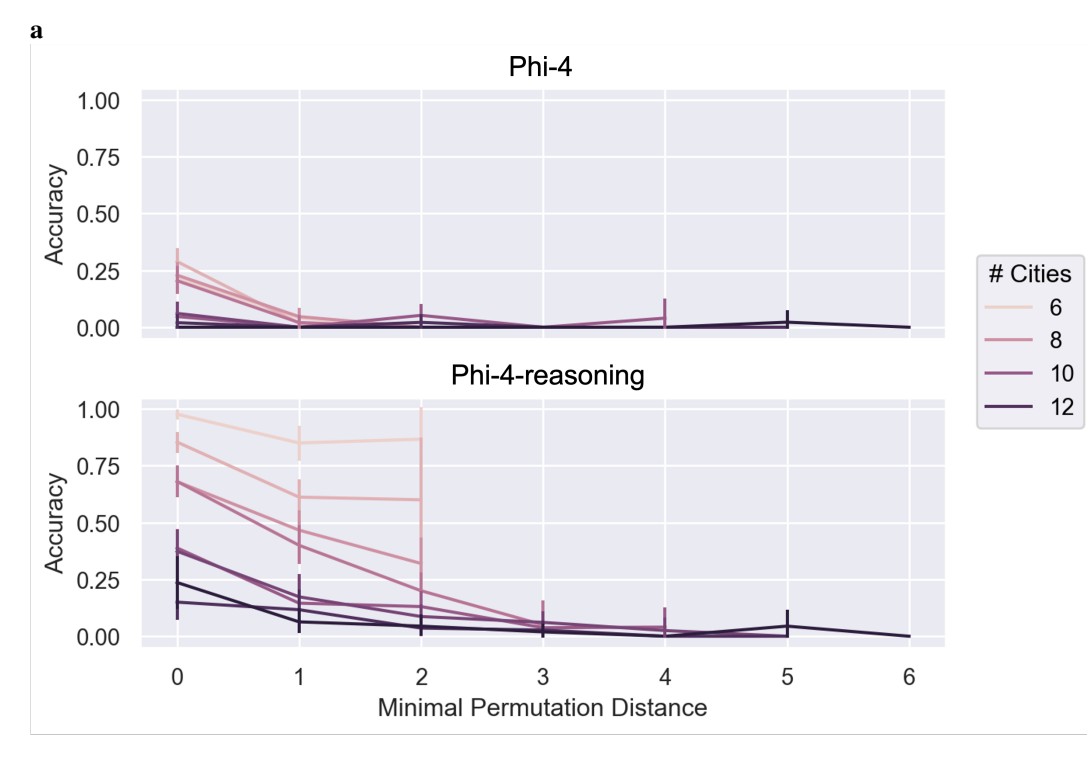

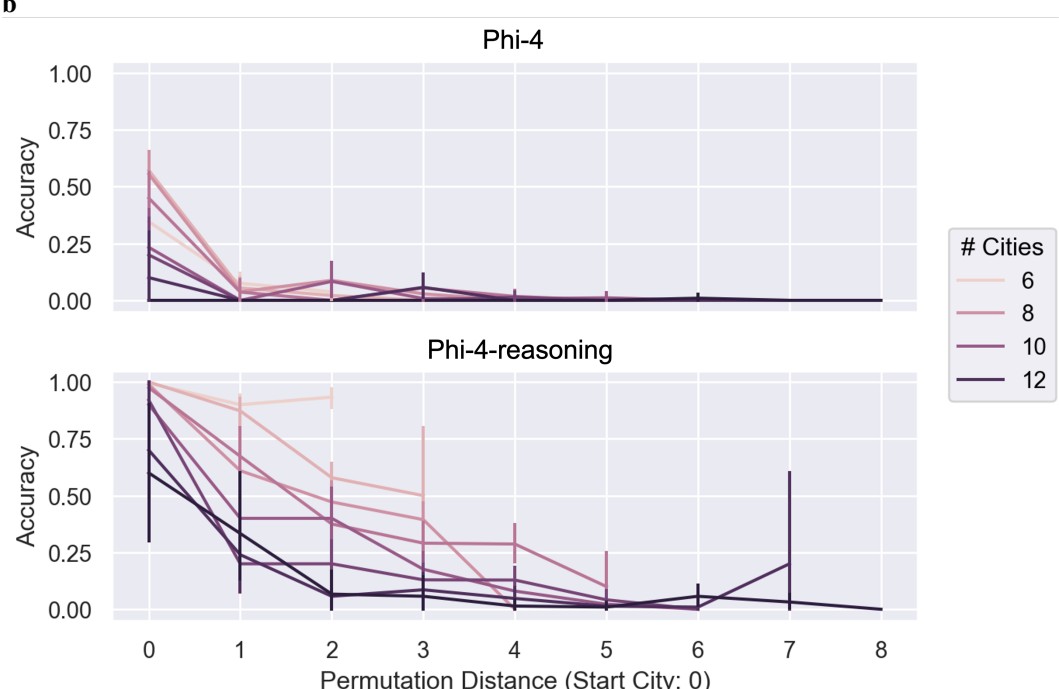

Figure 12: **a** The minimal NN-optimal edit distances with any city as the starting point. **b** The minimal NN-optimal edit distance using city 0 as the starting point. We observed a strong trend between edit distance and accuracy in **b**, even when controlling for the number of cities (and a somewhat weaker trend in **a**). This trend was especially strong for Phi-4-Reasoning, suggesting this model may effectively implement the nearest neighbor heuristic beginning at city 0.

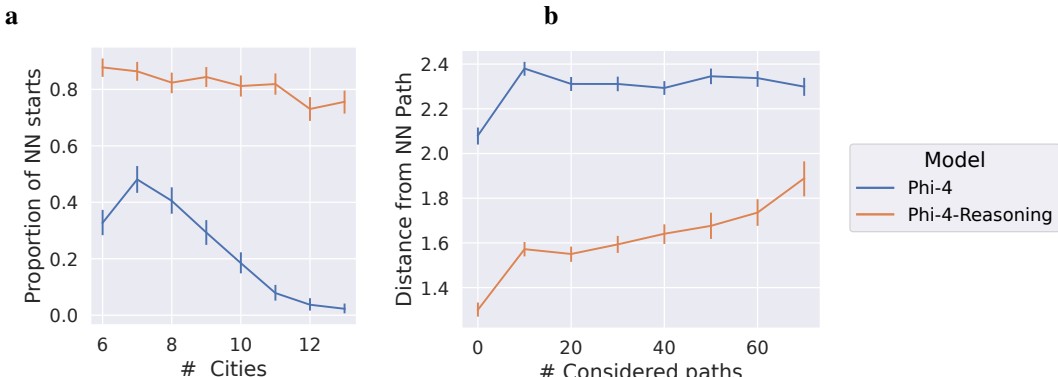

Figure 13: **a** We found that Phi-4 reasoning begins with the NN heuristic much more frequently than Phi-4-Base, especially when >8 cities are present. **b** Paths generated by Phi-4-Reasoning tend to have much lower distance on average from NN paths than those generated by Phi-4. Additionally, in Phi-4-Reasoning edit distance is lowest among the initial paths and increases as more candidate paths are selected.

