# OpenReview forum: "Algorithmic Primitives and Compositional Geometry of Reasoning in Language Models"
_ICLR.cc/2026/Conference — Submitted to ICLR 2026_

### Official Review · Reviewer_7Ty5 · 2025-10-26

**Soundness:** 3
**Presentation:** 2
**Contribution:** 3
**Rating:** 6
**Confidence:** 4

**Summary:**

This paper proposes a framework for identifying and steering algorithmic primitives in LLMs by combining clustering of latent activations with residual-stream vector interventions. Using tasks such as TSP, 3SAT, AIME, and graph navigation on models like Phi-4, Phi-4-Reasoning, and Llama-3-8B, the study shows that these primitives compose algebraically and transfer across tasks, revealing a compositional geometry of reasoning.

**Strengths:**

The paper presents a novel and thought-provoking framework for understanding the internal behaviors of LLMs.

The experiments cover multiple models and datasets, which provides credible support for the conclusions.

The analysis of reasoning-step transfer and compositional primitives is particularly interesting and offers new insights into LLM reasoning mechanisms.

**Weaknesses:**

The presentation and methodological details could be improved. For example, the paper lacks clarity regarding clustering hyperparameters and the statistical evaluation metrics, as noted in the questions below.

In addition, since the paper introduces several new concepts and definitions, it would be helpful to include a summary table that clearly defines all key terms included examples for better readability.

If these concerns are addressed, I would consider raising my score.

**Questions:**

In Section 3, what does the variable x specifically refer to?

How are candidate vectors selected for clustering? Please provide more detail on hyperparameter settings.
What are the quantitative results of clustering? Have you evaluated the quality using metrics such as ARI or NMI?

Did you observe differences in clustering behavior across layers? If so, how do these relate to the model’s reasoning hierarchy?

In Section 5, how is the “higher reward node” defined? What is the reference or baseline used for the comparison?

---

> ### Author Response · Authors · 2025-11-21
>
> We would like to thank the reviewer for the detailed feedback and helpful suggestions.
>
> > The presentation and methodological details could be improved. For example, the paper lacks clarity regarding clustering hyperparameters and the statistical evaluation metrics, as noted in the questions below.
>
> Thank you for this comment. We have clarified the methodological details, including clustering hyperparameters and statistical evaluation metrics, and provide additional explanation in the main text as well as in our responses below.
>
> > In addition, since the paper introduces several new concepts and definitions, it would be helpful to include a summary table that clearly defines all key terms included examples for better readability.
> > In Section 3, what does the variable x specifically refer to?
>
> That’s a great idea, we have now added a summary table of our key concepts alongside examples (Table 1). Please let us know if you have any further suggestions regarding this. Variable $x$ refers to an input the model is processing, which in the context of our work refers to a system and user prompt and the model’s response; we have now added this clarification also in Section 4.1 (l. 161).
>
> > How are candidate vectors selected for clustering?
>
> We now introduce a metric for the expressivity of a given cluster, measuring the number of unique tokens expressed by a cluster divided by the radius of that cluster. “Cluster Expressivity" measures the number of unique tokens expressed by a cluster, divided by the radius of that cluster in latent space (see Appendix A.7). We compute the median expressivity across all responses (the last column of the table). We find that primitives with high expressivity (expression density) often correspond to better-defined operations with rich vocabulary, with multiple valid ways to express the same operation. We take them as more likely to be "true" primitives with clear algorithmic meaning. Low expressivity can correspond to either vague or poorly defined clusters (maybe not real primitives), or conversely, operations that are so specific there are few ways to express them.
>
> This metric can be used as an automated criterion for the choice of primitive vectors: in particular, five of our six chosen primitives were among the ten clusters with highest expressivity. The other clusters with high associated expressivity had less distinct algorithmic roles; for example, cluster 18 associated with tokens covering general natural language, e.g. “check if we can get less”, “We can try to use lower bounds” or “One promising approach is to start”, that are used in the chain of thought process and provide a link between other more clearly defined primitives. We therefore picked a set of primitives that was representative of the overall algorithmic process for solving TSP. We are happy to run additional steering evaluations on the remaining clusters with high expressivity, however.
>
> > Have you evaluated the quality using metrics such as ARI or NMI?
>
> We note that while these metrics are often used to compare clusters to ground truth categories, there are no ground truth categories for these different algorithmic primitives in our case. However, we now use the adjusted Rand index and the adjusted normalized mutual information score to determine the degree of agreement between our clustering models trained with five responses and a set of clustering models trained with fifty responses (Fig. 11c). Please see more details below.

---

> ### Author Response · Authors · 2025-11-21
>
> > Did you observe differences in clustering behavior across layers? If so, how do these relate to the model’s reasoning hierarchy?
>
> > Please provide more detail on hyperparameter settings. What are the quantitative results of clustering?
>
> 1. To evaluate robustness to our choice of layers, we first repeated the clustering analysis for all layers in both Phi4 and Phi4-R for TSP (Traveling Salesperson). We then compared the frequency of each layer’s clusters showing up both across responses within a model, and across the responses of the two models. These dissimilarity matrices are displayed in the new appendix Fig. 9, a. To further evaluate uniquely informative patterns across the layers, we compared pair-wise similarity across layer-wise dissimilarity matrices, see new appendix Fig. 9, b.  Together, these analyses revealed broad patterns in the effect of layers: after the first layers, there was a large highly correlated cohort in the intermediate layers, especially in 10-25, centered on layers 17, and a smaller cohort of highly correlated layer layers, ranging from 30-40. We observed, first, that intermediate layers captured more aspects of the reasoning hierarchy than the later layers, and second, that there was more agreement among the models in the middle layers while the later models highlighted their differences. Together, these findings clearly justify layer 17 as a suitable candidate for clustering across models and responses. This is consistent with the findings in the main Figure 3a, b, where the middle layers are the most affected by our causal patching of algorithmic primitive vectors.
>
> 2. To evaluate robustness to our choice of k, we conducted a similar analysis, this time keeping the layer fixed (17) and varying K by 5, 30, 50, and 100. As described above, we first computed dissimilarity matrices for frequency of cluster occurrences in within-model and cross-model responses per K (new appendix Fig. 10a), and then computed the correlation across these matrices (new appendix Fig. 10b). Applied to 100 responses to TSP per model (Phi-4 and Phi-4R), this analysis revealed high correlations among K=30, 50, and 100, with the highest Pearson correlation of 0.98 between K=50 and 100.
> Next, we computed normalized inertia, or the sum of squared distance between each data point and its assigned cluster centroid for 4 tasks per K=5,10,...,100, with increments of 5 on the x axis (new appendix Fig. 10c). We then used the elbow method (Han et al. 2012) to identify around which K inertia dropped per task. Together, these findings revealed K=50 as the parsimonious and functionally appropriate choice, offering a reasonable tradeoff between explaining variance in the latent space and parsimony in the potential number of algorithmic primitives.
>
> 3. To evaluate robustness to sampling choice, we fixed the layer (17) and K=50, and evaluated the robustness of clustering to different numbers of responses, between 1 and 50. First, we found very high correlation between the identified dissimilarity matrices across varying numbers of responses with a minimum Pearson’s correlation of 0.96 for using 1 response, and a correlation of 0.9945 between 5 and 25 or 50 responses (new appendix Fig. 11 a and b). We then measured the consistency of assigned clusters across a varying number of responses with 2 measures: adjusted mutual information (MI) score and adjusted Rand score (new appendix Fig. 11 c), revealing high consistency for 5, 25, and 50 responses. Taken together these findings show robustness of assigned clusters to varying numbers of responses included in the analysis.
>
> We hope that these extensive sets of analysis address the reviewer’s excellent question. We describe these ablations in the newly added Appendix E.
>
> > In Section 5, how is the “higher reward node” defined? What is the reference or baseline used for the comparison?
>
> Reward is provided by a number between 1 and 100, e.g. given the input “rewards=[M:100 vs S:44]” the model should output “M”. We now clarify this in line 439. As a baseline, we use zero-shot evaluation (ZS), where the input is provided without any further context.
>
> We hope that our responses and revisions effectively addressed your concerns raised and thank you again for your helpful review. We would be happy to provide any additional clarifications and continue the discussion.

---

> > ### Comment · Reviewer_7Ty5 · 2025-11-25
> >
> > Thanks for your response. My concern has been addressed. I would like to increase the score.

---

### Official Review · Reviewer_NEzR · 2025-10-30

**Soundness:** 2
**Presentation:** 1
**Contribution:** 3
**Rating:** 2
**Confidence:** 4

**Summary:**

The paper proposes “Primitive Vectors” (PVs): directions in the residual stream extracted from attention head activity on self-generated reasoning traces, which can be injected at a target layer to steer behavior. The authors claim PVs correspond to reusable “algorithmic primitives” and that linear combinations compose these primitives.

**Strengths:**

1. **Mechanistic angle on reasoning.** Treats skills as additive residual directions with a tunable strength (α), connecting to activation patching / feature-vector literature.


2. **Compositional tests.** Evaluates whether sums/differences of PVs yield composed behaviors rather than single-feature nudges.

**Weaknesses:**

1. **Presentation gaps impede assessment.** Plots frequently lack axis labels and captions. Most importantly, there is no clear, self-contained equation in the main text for how PVs are computed. Line 191 points to §4.2; §4.2 (line 238) punts to the appendix; the appendix has no concrete definition. This is a blocker for reproducibility and for understanding what is actually averaged/normalized where.


2. **Heuristic choices are under-justified.** Why K-Means for potentially high-dimensional representations? Why (k=50)? Why layer 17? No sensitivity or ablations are provided to justify these design choices.


3. **Ad-hoc functionality assignment.** PV identities are hand-picked from 6 of 50 clusters after manual inspection. That feels brittle and hard to reproduce without objective criteria.


4. **Alternative explanation (bag-of-words steering).** Line 218 suggests K-Means is run over (h_17) for tokens in self-generated reasoning traces. If so, clusters may mostly track lexical/discourse cues (“alternatively”, “thus”, parentheses/digits) rather than procedure. The resulting average vector could simply boost a bag of tokens, which in turn nudges the next steps of a “reasoning” template. This also explains linear add/subtract as logit mass addition/suppression—not necessarily algorithmic composition. Suggest to add controlled experiments (e.g., unembedding alignment) to show this reflects a latent procedure rather than BOW steering.

The core idea is interesting, but the paper is hard to follow and omits key methodological details. With a precise PV definition and controls that rule out BOW steering, I could see this moving to 4 or better.

**Questions:**

1. Please provide a precise mathematical definition of PV construction (token set, head set, centering/normalization, layer choice, averaging domain, and scaling).


2. How sensitive are results to (k), distance metric, layer index, and α? Any reason to expect stability across model sizes?


3. For TSP, do the 6 selected clusters share any statistics that suggest an automated criterion (e.g., size, radius, density)?


Typos/Minor:

1. Line 784: broken appendix reference/link.

2. Please add axis labels and full captions to all figures.

---

> ### Author Response · Authors · 2025-11-21
>
> Thank you very much for your helpful review.
>
> > Presentation gaps impede assessment. Plots frequently lack axis labels and captions. Most importantly, there is no clear, self-contained equation in the main text for how PVs are computed. Line 191 points to §4.2; §4.2 (line 238) punts to the appendix; the appendix has no concrete definition. This is a blocker for reproducibility and for understanding what is actually averaged/normalized where.
>
> Thank you for pointing this out. We are fully committed to improving readability and reproducibility, and have now clarified labels and provide a clear definition of primitive vector computation (see Section B.6). We further include a self-contained paragraph in the main text that precisely defines how PVs are computed in Section 4.2 (l. 234-244, equation (1)), along with a brief step-by-step description and streamlined pointers to this definition. Please let us know if you have any further suggestions regarding this. We have copied our addition to the main section for your convenience below:
>
> We extract attention head activations (projected back into the residual dimension) for each response $j$, $Z^{(j)} \in \mathbb{R}^{T_j \times H \times L \times D}$, where $T_j$ is the number of tokens of response $j$, $H$ is the number of attention heads per layer, $L$ is the number of layers, and $D$ is the residual dimension. Denoting the set of tokens that are in a particular cluster $c$ by $T_j[c] \subseteq \\{1, \dotsc, T_j\\}$, we then compute the average attention head activations:
>
> $
> \overline{Z}[c] := \frac{1}{\sum_{j=1}^n |T_j[c]|} \sum_{j \in T_j[c]} Z^{(j)} \in \mathbb{R}^{L \times H \times D}
> $
>
> We then average these attention head activations over the top $k = 35$ attention heads that reliably carry out in-context learning functions (following the definition in Todd et al. (2024), see Appendix B.6 for further details).
>
> > Alternative explanation (bag-of-words steering). Line 218 suggests K-Means is run over (h_17) for tokens in self-generated reasoning traces. If so, clusters may mostly track lexical/discourse cues (“alternatively”, “thus”, parentheses/digits) rather than procedure. The resulting average vector could simply boost a bag of tokens, which in turn nudges the next steps of a “reasoning” template. This also explains linear add/subtract as logit mass addition/suppression—not necessarily algorithmic composition. Suggest to add controlled experiments (e.g., unembedding alignment) to show this reflects a latent procedure rather than BOW steering.
>
> Thank you for this insightful comment; it prompted us to perform a dedicated control experiment to differentiate an actual latent procedure from a simple bag-of-words explanation. We reasoned that if the primitive vector simply boosts a bag of tokens, the algorithmic expressions should be evoked with similar strength by simply adding this vector to the end of the residual stream, directly before the unembedding. (This is what we understood your suggestion to be, but please let us know if you had something else in mind.) When controlling for this (by subtracting the maximal effect obtained by injecting the primitive vectors at the end of the residual stream), we still observe the same effects, suggesting that they reflect a latent procedure (see Table 3 and Appendix D.1 in the paper, copied below for your convenience).
>
> | Method                  | %NN paths ↑  | #Paths ↑     | Dist. comp. ↓ | Final ans. ↓ | #Verif. ↑   | #Comp. ↑     |
> |-------------------------|--------------|-------------|---------------|--------------|------------|-------------|
> | `nearest_neighbor`      | **+33.2%**   | _+47.3%_    | _-42.2%_      | -18.6%       | +9.8%      | +141.5%     |
> | `generate_new_path`     | +0.0%        | **+73.9%**  | -19.7%        | -0.3%        | -213.0%    | -7.3%       |
> | `compute_distance`      | +10.1%       | +33.7%      | **-44.8%**    | _-50.5%_     | +128.3%    | +103.7%     |
> | `produce_final_answer`  | _+12.8%_     | +34.0%      | -21.2%        | **-63.5%**   | +47.8%     | -1.2%       |
> | `compare_or_verify`     | +7.9%        | +38.5%      | -38.0%        | -20.2%       | **+559.8%**| **+1059.8%**|
> | `compare_or_verify_2`   | +5.9%        | +0.8%       | -24.3%        | -24.7%       | _+516.3%_  | _+290.2%_  |
>
> In summary, these results confirm that the observed effects are not merely demonstrated through a bag-of-words effect, rather reflecting an underlying latent procedure. As we think this is an important differentiation of our approach compared to related works on analyzing the procedure behind more controlled token-specific strategies, e.g. linebreaks (https://transformer-circuits.pub/2025/linebreaks/index.html), we will include a dedicated discussion to this in Section 5.

---

> > ### Author Response · Authors · 2025-11-21
> >
> > > Heuristic choices are under-justified. Why K-Means for potentially high-dimensional representations? Why (k=50)? Why layer 17? No sensitivity or ablations are provided to justify these design choices.
> >
> > 1. Our choice was informed by the fact that recent investigations in large language models determined that k-means provided suitable results on discovering concepts in latent LLM representations  (e.g. https://aclanthology.org/2024.eacl-long.48.pdf, https://www.sciencedirect.com/science/article/pii/S2666307424000482, https://openreview.net/forum?id=btJUnAPQ7j). We now clarify this in l. 216-218. In practice, we found that k-means yields well-defined, interpretable clusters in the original high-dimensional space, which is further demonstrated by our automatic labeling of all clusters (see also our responses below), allowing us to extract meaningful primitive vectors.
> >
> > 2. To evaluate robustness to our choice of layers, we first repeated the clustering analysis for all layers in both Phi4 and Phi4-R for TSP (Traveling Salesperson). We then compared the frequency of each layer’s clusters showing up both across responses within a model, and across the responses of the two models. These dissimilarity matrices are displayed in the new appendix Fig. 9, a. To further evaluate uniquely informative patterns across the layers, we compared pair-wise similarity across layer-wise dissimilarity matrices, see new appendix Fig. 9, b.  Together, these analyses revealed broad patterns in the effect of layers: after the first layers, there was a large highly correlated cohort in the intermediate layers, especially in 10-25, centered on layers 17, and a smaller cohort of highly correlated layer layers, ranging from 30-40. We observed, first, that intermediate layers captured more aspects of the reasoning hierarchy than the later layers, and second, that there was more agreement among the models in the middle layers while the later models highlighted their differences. Together, these findings clearly justify layer 17 as a suitable candidate for clustering across models and responses. This is consistent with the findings in the main Figure 3a, b, where the middle layers are the most affected by our causal patching of algorithmic primitive vectors.
> >
> > 3. To evaluate robustness to our choice of k, we conducted a similar analysis, this time keeping the layer fixed (17) and varying K by 5, 30, 50, and 100. As described above, we first computed dissimilarity matrices for frequency of cluster occurrences in within-model and cross-model responses per K (new appendix Fig. 10a), and then computed the correlation across these matrices (new appendix Fig. 10b). Applied to 100 responses to TSP per model (Phi-4 and Phi-4R), this analysis revealed high correlations among K=30, 50, and 100, with the highest Pearson correlation of 0.98 between K=50 and 100.
> > Next, we computed normalized inertia, or the sum of squared distance between each data point and its assigned cluster centroid for 4 tasks per K=5,10,...,100, with increments of 5 on the x axis (new appendix Fig. 10c). We then used the elbow method (Han et al. 2012) to identify around which K inertia dropped per task. Together, these findings revealed K=50 as the parsimonious and functionally appropriate choice, offering a reasonable tradeoff between explaining variance in the latent space and parsimony in the potential number of algorithmic primitives.
> >
> > 4. To evaluate robustness to sampling choice, we fixed the layer (17) and K=50, and evaluated the robustness of clustering to different numbers of responses, between 1 and 50. First, we found very high correlation between the identified dissimilarity matrices across varying numbers of responses with a minimum Pearson’s correlation of 0.96 for using 1 response, and a correlation of 0.9945 between 5 and 25 or 50 responses (new appendix Fig. 11 a and b). We then measured the consistency of assigned clusters across a varying number of responses with 2 measures: adjusted mutual information (MI) score and adjusted Rand score (new appendix Fig. 11 c), revealing high consistency for 5, 25, and 50 responses. Taken together these findings show robustness of assigned clusters to varying numbers of responses included in the analysis.
> >
> > We hope that these extensive sets of analysis address the reviewer’s excellent question. We describe these ablations in the newly added Appendix E.

---

> > > ### Author Response · Authors · 2025-11-21
> > >
> > > > Ad-hoc functionality assignment. PV identities are hand-picked from 6 of 50 clusters after manual inspection. That feels brittle and hard to reproduce without objective criteria.
> > >
> > > We have now included a novel automated pipeline for labeling primitive clusters using LLMs (description in Appendix F) and its results. This allows for (i) the automated labeling of identified clusters, and (ii) the verification of manually assigned labels. We found that the automatically extracted labels agree with the manually assigned roles (see Table below, annotations for all fifty clusters are added to the Appendix, Table 5).
> > >
> > > | Manual Label            | Automated (Token-centric)                               | Automated (Context-centric)                                           |
> > > |-------------------------|----------------------------------------------------------|------------------------------------------------------------------------|
> > > | `produce_final_answer`  | Output formatting with <final_answer> tag               | Output formatting with <final_answer> tag.                             |
> > > | `compare_or_verify_2`   | Route exploration and modification suggestions           | Route evaluation and improvement through swaps and recalculations.     |
> > > | `nearest_neighbor`      | "Selecting best path based on edge weights"              | "Selecting best path based on edge costs."                             |
> > > | `compare_or_verify`     | Proposed alternative routes                              | Route optimization and evaluation in TSP solutions.                    |
> > > | `compute_distance`      | Summation and accumulation of path costs                 | Addition and summation of route distances.                             |
> > > | `generate_new_path`     | Proposed routes for alternative solutions                | Route exploration and evaluation for optimization.                     |
> > >
> > >
> > > Moreover, this pipeline can be integrated in an automated end-to-end implementation of our framework, using a novel metric we propose to quantify the expressivity of each cluster. “Cluster Expressivity" measures the number of unique tokens expressed by a cluster, divided by the radius of that cluster in latent space (formally defined in the added Appendix A.7). We compute the median expressivity across all responses (the last column of the table). We find that primitives with high expressivity (expression density) often correspond to better-defined operations with rich vocabulary, with multiple valid ways to express the same operation. We take them as more likely to be "true" primitives with clear algorithmic meaning. Low expressivity can correspond to either vague or poorly defined clusters (maybe not real primitives), or conversely, operations that are so specific there are few ways to express them. Notably, we find that our manual labels are distinguished by very high expressivity (see Table 5).
> > >
> > > To evaluate robustness of our automated pipeline to different hyperparameters, we further analyzed the automatic labels using k=100 clusters (see new appendix F.2). We found that the most distinguishing clusters between Phi-4 and Phi-4-Reasoning (with the largest $\chi^2$-difference) had reflected the differences in algorithmic approach we had previously identified (Table 6). In particular, the top cluster in Phi-4-Reasoning expressed a nearest-neighbor approach, whereas the top cluster in Phi-4 expressed a brute-force approach. Crucially, we did not have to manually analyze these clusters at all and were simply able to draw these conclusions from the automated labelling pipeline. Overall, this shows that 1) our hand-identified algorithmic primitives are robust to both automation and (reasonable) choices of hyperparameters, and 2) our novel labeling pipeline enables automated end-to-end clustering and primitive identification.
> > >
> > > We hope that our automated pipeline for labeling clusters together with the expressivity metric guiding this choice of primitives adequately addresses the reviewer’s questions.

---

> > > > ### Author Response · Authors · 2025-11-21
> > > >
> > > > > Please provide a precise mathematical definition of PV construction (token set, head set, centering/normalization, layer choice, averaging domain, and scaling).
> > > >
> > > > We have now added a precise definition to Section B.6, summarizing this definition with the key equation in Section 4.2 (see also our response above). We reproduce our response from the section for your convenience below:
> > > >
> > > > To extract primitive vectors we first compute the average indirect effect (AIE), as described in Section 2.3 of Todd et al. (2024), averaging over the same set of six in-context learning tasks (antonym, capitalize, country-capital, english-french, present-past, singular-plural). We then pick the 35 attention heads with the largest AIE, denoting them by
> > > > $\mathcal{A} \subseteq \\{(l,h)\mid l = 1,\dotsc,40,\; h = 1,\dotsc,40\\}$,
> > > > where $l$ denotes the layer of the corresponding attention head and $h$ denotes its particular index.
> > > >
> > > > We consider a set of $n = 200$ responses in total (100 responses from Phi-4 and 100 responses from Phi-4-Reasoning). For each response $j = 1,\dotsc,n$, we extract the activities in every attention head on this response,
> > > > $Z^{(j)} \in \mathbb{R}^{T_j \times L \times H \times D}$,
> > > > where $L = 40$ is the number of layers, $H = 40$ is the number of attention heads, $T_j$ is the number of tokens in response $j$, and $D$ is the residual dimension ($D = 5420$).
> > > >
> > > > To compute $Z^{(j)}$, we project the attention-head activations back into the residual dimension using the projection weights following those attention heads (the same method as Todd et al. (2024)). Like Todd et al. (2024), we do not apply any further scaling or normalization. For a given cluster $c$, we denote all tokens that belong to the cluster by
> > > > $T_j[c] \subseteq \\{1,\dotsc,T_j\\}$.
> > > > We then compute the average activity in each attention head over all those tokens:
> > > >
> > > > $$
> > > > \overline{Z}[c]
> > > > :=
> > > > \frac{1}{\sum_{j=1}^n |T_j[c]|}
> > > > \sum_{j \in T_j[c]} Z^{(j)}
> > > > \in \mathbb{R}^{L \times H \times D}.
> > > > $$
> > > >
> > > > Finally, we compute the primitive vector $v^{(p)}[c]$ corresponding to this cluster as the average across the 35 attention heads with the largest AIE:
> > > >
> > > > $$v^{(p)}[c]=\frac{1}{|\mathcal{A}|}
> > > > \sum_{l,h \in \mathcal{A}}
> > > > \overline{Z}_{lh}
> > > > \in \mathbb{R}^{D}.
> > > > $$
> > > >
> > > > > How sensitive are results to (k), distance metric, layer index, and α? Any reason to expect stability across model sizes?
> > > >
> > > > We have now included ablations for the choice of k and layer, as we detail above. Regarding $\alpha$, we find that maximal expression of a given primitive often arises when it is injected with an intermediate magnitude (see e.g. Fig. 3 b,c). We chose Euclidean distance as a standard choice for k-means clustering; if you think there would be important distance metrics to compare our findings to, we would be happy to run these additional experiments.
> > > >
> > > > > For TSP, do the 6 selected clusters share any statistics that suggest an automated criterion (e.g., size, radius, density)?
> > > >
> > > > As mentioned above, we now introduce a metric for the expressivity of a given cluster, measuring the number of unique tokens expressed by a cluster divided by the radius of that cluster. Note that this metric was inspired by your suggestion and we sincerely appreciate your helpful question. This metric can be used as an automated criterion for the choice of primitive vectors: in particular, five of our six chosen primitives were among the ten clusters with highest expressivity. The other clusters with high associated expressivity had less distinct algorithmic roles; for example, cluster 18 associated with tokens covering general natural language, e.g. “check if we can get less”, “We can try to use lower bounds” or “One promising approach is to start”, that are used in the chain of thought process and provide a link between other more clearly defined primitives. We therefore picked a set of primitives that was representative of the overall algorithmic process for solving TSP. We are happy to run additional steering evaluations on the remaining clusters with high expressivity, however.
> > > >
> > > > > Typos/Minor:
> > > > Line 784: broken appendix reference/link.
> > > > Please add axis labels and full captions to all figures.
> > > >
> > > > Thank you for pointing this out, it’s fixed in the revised version.
> > > >
> > > > We hope that our responses and revisions effectively addressed your concerns raised and thank you again for your helpful review. We would be happy to provide any additional clarifications and continue the discussion.

---

> ### Comment · Reviewer_NEzR · 2025-11-26
>
> Thank you for the detailed rebuttal and revisions. My main concerns about the missing PV definition, the heuristic choices (layer / k / sampling), and the ad-hoc cluster selection are now well addressed by the new analyses and clarifications. I appreciate the amount of additional work that went into this; I will increase my score to 6.
>
> To clarify my earlier comment on the bag-of-words alternative: I had in mind something closer to what the original Function Vectors paper did—**decode** each PV directly into words, and inspect the top-aligned tokens. One can then optionally reconstruct an approximate PV from the top-k tokens and check whether **the same function-modifying behavior remains**. Concretely, this could be done (in its simplest form) by taking the primitive vector $(v^{(p)})$, computing $(u = U v^{(p)})$ where $(U)$ is the unembedding (vocab × d), and ranking tokens by $(u)$ (or by their logit change under a small injection).

---

> > ### Author Response · Authors · 2025-12-03
> >
> > Thank you for your positive response; we are glad that we were able to address your concerns and are grateful that you increased your score.
> >
> > We appreciate the clarification regarding the bag-of-words alternative. Our aim in including the original control was to test whether the observed effects could be explained by simply shifting the logits toward a set of tokens associated with a primitive vector (i.e. $u=Uv^{(p)}$), rather than by a latent procedure. Because injecting the primitive vector at the end of the residual stream directly boosts the logits for its top-aligned tokens, if a bag-of-words mechanism were driving the behavior, this intervention should approximate the effects of injections into intermediate layers. We observed that this was not the case (see Table 3), indicating that the effect of the primitive vectors is better explained by an underlying latent procedure.
> >
> > In line with your suggestion, we now also evaluate whether the primitive vector can be inferred from its top decoded tokens, implementing the analysis pursued in Todd et al. Specifically, we reconstructed the vector from its top-100 decoded tokens, as detailed in the newly added Appendix D.3. We found that in almost all cases, the effect of this reconstructed vector on behavior was smaller than that of the original primitive vector, with a notable exception being the effect of compare_or_verify on the number of comparisons (see the table in Appendix D.3). This indicates that primitive vectors usually contain some information beyond what can be inferred from their top decoded tokens. Thank you very much for your suggestion and thank you again for your helpful review!

---

### Official Review · Reviewer_EaF2 · 2025-11-01

**Soundness:** 3
**Presentation:** 3
**Contribution:** 3
**Rating:** 6
**Confidence:** 3

**Summary:**

This paper  proposes a framework for finding algorithmic primitive vectors in LLMs. Primitive vectors are vectors that are found by averaging activations in LLMs that are associated with a unique algorithm building block. Authors use clustering methods to identify tokens with similar activation patterns, average activations within a cluster and then in a bottom-up approach associate each primitive with an algorithmic behavior. Once found, these vectors can be used to steer LLMs towards relying more on those algortithmic building blocks. In the case of this paper, authors study algorithmic primitives in the Phi4 and Phi4 reasoning models and focus on reasoning tasks. Findings include the observation that reasoning fine-tuning leads to more general resolution algorithmic building block. Also: "We find that algorithmic primitives exhibit geometric regularities through compositional operations like addition, subtraction, multiplication,and scalar modulation. The layer and magnitude of injection shape the output expression of the primitive (Fig. 2). We show cross-task transfer and generalizability of primitive vectors: spatial reasoning and verification primitives extracted from AIME successfully transfer to TSP. This transferability hints at potentially universal algorithmic building blocks underlying diverse reasoning capabilities.

**Strengths:**

- Form: paper is well-written, easy to follow, polished
- Relevance: as pointed out by the authors, the question investigated by the authors participates to the general conversation about whether LLMs reason or memorize. Through the proposed framework and findings, this work suggests that general algorithmic building blocks are used by LLMs to solve tasks requiring reasoning and further highlight the geometric structure in the representation space of LLMs. The topic is hence relevant to the community.
- Simplicity of the proposed method: the framework proposed by authors is straightforward, driven by first principles, easy to translate (task and model agnostic), hence its application to other settings seems straightforward

**Weaknesses:**

- Missing ablations to support robustness of the proposed method: the authors propose a framework that can be used to understand pre-trained models. While straightforward, the method relies on the application of a clustering method which itself requires a choice of a number of clusters and a layer. How are those chosen? How sensitive is the analysis to these choices? how robust are these hyperparameters across models? There are limited explanations regarding those choices which seem important to be able to sell a transferable framework. How many of those clusters were not found to be linked with a clear algorithmic trace?
- Scope of empirical evidence: the findings are insightful notably the observations that Phi4 reasoning relies on primitive associated with general and reusable algorithmic building blocks. However, I think the scope of the experiments should be extend to more models in order for the analysis to be impactful.
- Questions about Impact: While the findings are relevant to the community and the proposed method is straightforward, I have concerns about the overall impact of the work—particularly regarding how well the approach aligns with the actual internal computations of LLMs. The method relies on clustering to extract primitive vectors associated with algorithmic traces. However, the results in Table 1 indicate that steering the model using a single primitive affects multiple algorithmic components, suggesting that these behaviors are not disentangled. Consequently, I question the significance of the proposed approach and its findings, as the primitives lack clear independence, and the paper does not provide a comprehensive analysis of their dependencies or a causal model linking them.

**Questions:**

See weaknesses

---

> ### Author Response · Authors · 2025-11-21
>
> Thank you very much for your helpful review.
>
> > Missing ablations to support robustness of the proposed method: the authors propose a framework that can be used to understand pre-trained models. While straightforward, the method relies on the application of a clustering method which itself requires a choice of a number of clusters and a layer. How are those chosen? How sensitive is the analysis to these choices? how robust are these hyperparameters across models? There are limited explanations regarding those choices which seem important to be able to sell a transferable framework.
>
> 1. To evaluate robustness to our choice of layers, we first repeated the clustering analysis for all layers in both Phi4 and Phi4-R for TSP (Traveling Salesperson). We then compared the frequency of each layer’s clusters showing up both across responses within a model, and across the responses of the two models. These dissimilarity matrices are displayed in the new appendix Fig. 9, a. To further evaluate uniquely informative patterns across the layers, we compared pair-wise similarity across layer-wise dissimilarity matrices, see new appendix Fig. 9, b.  Together, these analyses revealed broad patterns in the effect of layers: after the first layers, there was a large highly correlated cohort in the intermediate layers, especially in 10-25, centered on layers 17, and a smaller cohort of highly correlated layer layers, ranging from 30-40. We observed, first, that intermediate layers captured more aspects of the reasoning hierarchy than the later layers, and second, that there was more agreement among the models in the middle layers while the later models highlighted their differences. Together, these findings clearly justify layer 17 as a suitable candidate for clustering across models and responses. This is consistent with the findings in the main Figure 3a, b, where the middle layers are the most affected by our causal patching of algorithmic primitive vectors.
>
> 2. To evaluate robustness to our choice of k, we conducted a similar analysis, this time keeping the layer fixed (17) and varying K by 5, 30, 50, and 100. As described above, we first computed dissimilarity matrices for frequency of cluster occurrences in within-model and cross-model responses per K (new appendix Fig. 10a), and then computed the correlation across these matrices (new appendix Fig. 10b). Applied to 100 responses to TSP per model (phi-4 and Phi-4R), this analysis revealed high correlations among K=30, 50, and 100, with the highest Pearson correlation of 0.98 between K=50 and 100.
> Next, we computed normalized inertia, or the sum of squared distance between each data point and its assigned cluster centroid for 4 tasks per K=5,10,...,100, with increments of 5 on the x axis (new appendix Fig. 10c). We then used the elbow method (Han et al. 2012) to identify around which K inertia dropped per task. Together, these findings revealed K=50 as the parsimonious and functionally appropriate choice, offering a reasonable tradeoff between explaining variance in the latent space and parsimony in the potential number of algorithmic primitives.
>
> > How many of those clusters were not found to be linked with a clear algorithmic trace?
>
> Thank you for this question. To broaden our analyses beyond the six investigated clusters in our paper, we extracted primitive labels through automatic labeling for all clusters (Table 5). We find that these overall result in algorithmically distinct roles in around 76% (38 out of 50 clusters)  of cases (see Appendix F.1), compared to 24% (12 out of 50 clusters) that are characterized by linguistic patterns or only very few tokens.
>
> We will add discussion on the different functional roles of primitives to the final version of our paper.
>
> > Scope of empirical evidence: the findings are insightful notably the observations that Phi4 reasoning relies on primitive associated with general and reusable algorithmic building blocks. However, I think the scope of the experiments should be extend to more models in order for the analysis to be impactful.
>
> Thank you for this comment. We agree that extending our experiments to additional models would further strengthen the impact of the work. In this revision, we prioritized ablations, control experiments, and the automated labeling pipeline, but we now note cross-model evaluation as an important direction for future research (l. 525).

---

> > ### Author Response · Authors · 2025-11-21
> >
> > > Questions about Impact: While the findings are relevant to the community and the proposed method is straightforward, I have concerns about the overall impact of the work—particularly regarding how well the approach aligns with the actual internal computations of LLMs. The method relies on clustering to extract primitive vectors associated with algorithmic traces. However, the results in Table 1 indicate that steering the model using a single primitive affects multiple algorithmic components, suggesting that these behaviors are not disentangled. Consequently, I question the significance of the proposed approach and its findings, as the primitives lack clear independence, and the paper does not provide a comprehensive analysis of their dependencies or a causal model linking them.
> >
> > We thank the reviewer for their thoughtful comments regarding the impact of our work and question regarding the disentanglement of primitives.
> >
> > The entanglement and context-dependency of multiple primitives could have a number of causes. First, entanglement may reflect how certain primitives often co-occur or are chained together in reasoning. For instance, Compare_or_verify primitives often occur after path_generation, so the entanglement of their representations may mirror their functional relationship. Thus, the entanglement may follow reasoning motif structure, where primitives involved in a recurring multi-step motif (sequence of multiple primitives) are likely to be more entangled. This suggests the context-dependency may capture meaningful algorithmic patterns, not just noise. Second, some primitives may have computational similarity. For instance, given the hierarchical nature of primitives, they may share a number of computational sub-primitives. Third, a given primitive such as Compare_or_verify evokes different sub-routines or sub-primitives depending on the context, e.g., it compares distances after generating paths or verifies the accuracy of distance computation. While the second point mostly concerns cross-primitive similarity as a source of entanglement, the third point regards context-dependency of the same primitive. We will add this explanation to the discussion section of the paper.
> >
> > In Table 2, each cell had its own optimized injection magnitude $\alpha$ and injection layer L (maximizing expression for that specific primitive-task combination). To further investigate the specificity of algorithmic primitives, we have now added an Appendix Table 4, where each row uses a single $\alpha$ and L (optimized for that row's primitive, then evaluated across all metrics in that row). We copy this table here for the reviewer's convenience:
> >
> > | Method                  | %NN paths ↑  | #Paths ↑     | Dist. comp. ↓ | Final ans. ↓ | #Verif. ↑   | #Comp. ↑     |
> > |-------------------------|--------------|-------------|---------------|--------------|------------|-------------|
> > | `nearest_neighbor`      | **+56.1%**   | -4.2%       | _-34.9%_      | +6.3%        | _+40.2%_   | -22.0%      |
> > | `generate_new_path`     | -41.4%       | **+143.9%** | _-76.0%_      | +85.3%       | -93.5%     | -82.9%      |
> > | `compute_distance`      | -100.0%      | -100.0%     | **-86.5%**    | +37.9%       | _+35.9%_   | _+79.3%_   |
> > | `produce_final_answer`  | -100.0%      | -95.2%      | +283.8%       | **-81.5%**   | -48.9%     | -84.1%      |
> > | `compare_or_verify`     | -100.0%      | -95.5%      | +266.4%       | +85.5%       | _+568.5%_  | **+1103.7%**|
> > | `compare_or_verify_2`   | -100.0%      | -97.7%      | +280.2%       | +76.5%       | **+706.5%**| _+325.6%_  |
> >
> > Primitive vectors were substantially more disentangled in this analysis. The analysis reveals that the entanglement of some primitives seems to follow the sequential order of their usual expression. For instance, injecting nearest_neighbor increases the expression of its algorithmic successor, distance_computation, but injecting the successor doesn’t increase the expression of its predecessor (nearest_neighbor). This preliminary finding is consistent with the “Motif structure” hypothesis above, suggesting that the entanglement of two primitives may be, at least partially, due to them being commonly expressed in specific sequential algorithmic motifs, with specific ordering. Future dedicated studies are required to further investigate these hypotheses as well as the larger primitive-primitive relational motifs (see also Figure 2 c, d).
> >
> > We hope that our responses and revisions effectively addressed your concerns raised and thank you again for your helpful review. We would be happy to provide any additional clarifications and continue the discussion.

---

### Official Review · Reviewer_Tj9J · 2025-11-04

**Soundness:** 3
**Presentation:** 2
**Contribution:** 3
**Rating:** 6
**Confidence:** 2

**Summary:**

The paper proposes an “algorithmic primitives” framework to trace and steer multi-step reasoning in LLMs. It clusters internal activations to discover recurring primitives (e.g., generate_path, nearest_neighbor, compare_or_verify), maps them to reasoning traces, then derives primitive vectors (via averaged top attention-head outputs) that can be injected into residual streams to induce, suppress, and compose reasoning behaviors. Across TSP, 3-SAT, AIME, and graph navigation and across Phi-4, Phi-4-Reasoning, and Llama-3-8B. the authors show primitives exhibit a compositional geometry (addition, subtraction, scalar scaling), transfer across tasks, and differ systematically between base vs. reasoning-finetuned models (e.g., more verification/planning in Phi-4-Reasoning).

**Strengths:**

1. The problem this paper is exploring is interesting to me. Vector injection and subtraction produce clear, selective changes in reasoning hallmarks (e.g., verification spikes).

2. The expeirments demonstrate additive composition that induces a composite skill and cross-task transfer, suggesting shareable, reusable building blocks.

3. By mapping token traces, activation clusters, primitive vectors and layering in the temporal/meta-cluster dynamics, the work delivers a genuinely end-to-end interpretability account.

**Weaknesses:**

1. The use of layer 17, k=50, and small response batches risks making the identified primitives dependent on those choices and vulnerable to sampling noise.

2. The mapping from clusters to primitive labels is hand-interpreted from local text, and some clusters (e.g., compare_or_verify) appear context-dependent and multi-purpose.

**Questions:**

1. Can we have time complexity for this methods?

---

> ### Author Response · Authors · 2025-11-21
>
> Thank you very much for your helpful review.
>
> > The use of layer 17, k=50, and small response batches risks making the identified primitives dependent on those choices and vulnerable to sampling noise.
>
> 1. To evaluate robustness to our choice of layers, we first repeated the clustering analysis for all layers in both Phi4 and Phi4-R for TSP (Traveling Salesperson). We then compared the frequency of each layer’s clusters showing up both across responses within a model, and across the responses of the two models. These dissimilarity matrices are displayed in the new appendix Fig. 9, a. To further evaluate uniquely informative patterns across the layers, we compared pair-wise similarity across layer-wise dissimilarity matrices, see new appendix Fig. 9, b.  Together, these analyses revealed broad patterns in the effect of layers: after the first layers, there was a large highly correlated cohort in the intermediate layers, especially in 10-25, centered on layers 17, and a smaller cohort of highly correlated layer layers, ranging from 30-40. We observed, first, that intermediate layers captured more aspects of the reasoning hierarchy than the later layers, and second, that there was more agreement among the models in the middle layers while the later models highlighted their differences. Together, these findings clearly justify layer 17 as a suitable candidate for clustering across models and responses. This is consistent with the findings in the main Figure 3a, b, where the middle layers are the most affected by our causal patching of algorithmic primitive vectors.
>
> 2. To evaluate robustness to our choice of k, we conducted a similar analysis, this time keeping the layer fixed (17) and varying K by 5, 30, 50, and 100. As described above, we first computed dissimilarity matrices for frequency of cluster occurrences in within-model and cross-model responses per K (new appendix Fig. 10a), and then computed the correlation across these matrices (new appendix Fig. 10b). Applied to 100 responses to TSP per model (phi-4 and Phi-4R), this analysis revealed high correlations among K=30, 50, and 100, with the highest Pearson correlation of 0.98 between K=50 and 100.
> Next, we computed normalized inertia, or the sum of squared distance between each data point and its assigned cluster centroid for 4 tasks per K=5,10,...,100, with increments of 5 on the x axis (new appendix Fig. 10c). We then used the elbow method (Han et al. 2012) to identify around which K inertia dropped per task. Together, these findings revealed K=50 as the parsimonious and functionally appropriate choice, offering a reasonable tradeoff between explaining variance in the latent space and parsimony in the potential number of algorithmic primitives.
>
> 3. To evaluate robustness to sampling choice, we fixed the layer (17) and K=50, and evaluated the robustness of clustering to different numbers of responses, between 1 and 50. First, we found very high correlation between the identified dissimilarity matrices across varying numbers of responses with a minimum Pearson’s correlation of 0.96 for using 1 response, and a correlation of 0.9945 between 5 and 25 or 50 responses (new appendix Fig. 11 a and b). We then measured the consistency of assigned clusters across a varying number of responses with 2 measures: adjusted mutual information (MI) score and adjusted Rand score (new appendix Fig. 11 c), revealing high consistency for 5, 25, and 50 responses. Taken together these findings show robustness of assigned clusters to varying numbers of responses included in the analysis.
>
> We hope that these extensive sets of analysis address the reviewer’s excellent question. We describe these ablations in the newly added Appendix E.

---

> > ### Author Response · Authors · 2025-11-21
> >
> > > The mapping from clusters to primitive labels is hand-interpreted from local text
> >
> > Thank you for raising this important point. We have now included a novel automated pipeline for labeling primitive clusters using LLMs (description in Appendix F) and its results. This allows for (i) the automated labeling of identified clusters, and (ii) the verification of manually assigned labels. We found that the automatically extracted labels agree with the manually assigned roles (see Table below, annotations for all fifty clusters are added to the Appendix, Table 5). (Note that we suggest using the two provided labels in conjunction to infer the role of a particular cluster.)
> >
> > | Manual Label            | Automated (Token-centric)                               | Automated (Context-centric)                                           |
> > |-------------------------|----------------------------------------------------------|------------------------------------------------------------------------|
> > | `produce_final_answer`  | Output formatting with \<final_answer\> tag               | Output formatting with \<final_answer\> tag.                             |
> > | `compare_or_verify_2`   | Route exploration and modification suggestions           | Route evaluation and improvement through swaps and recalculations.     |
> > | `nearest_neighbor`      | "Selecting best path based on edge weights"              | "Selecting best path based on edge costs."                             |
> > | `compare_or_verify`     | Proposed alternative routes                              | Route optimization and evaluation in TSP solutions.                    |
> > | `compute_distance`      | Summation and accumulation of path costs                 | Addition and summation of route distances.                             |
> > | `generate_new_path`     | Proposed routes for alternative solutions                | Route exploration and evaluation for optimization.                     |
> >
> >
> > Moreover, this pipeline can be integrated in an automated end-to-end implementation of our framework, using a novel metric we propose to quantify the expressivity of each cluster. “Cluster Expressivity" measures the number of unique tokens expressed by a cluster, divided by the radius of that cluster in latent space (formally defined in the added Appendix A.7). We compute the median expressivity across all responses (the last column of the table). We find that primitives with high expressivity (expression density) often correspond to better-defined operations with rich vocabulary, with multiple valid ways to express the same operation. We take them as more likely to be "true" primitives with clear algorithmic meaning. Low expressivity can correspond to either vague or poorly defined clusters (maybe not real primitives), or conversely, operations that are so specific there are few ways to express them. Notably, we find that our manual labels are distinguished by very high expressivity (see Table 5).
> >
> > To evaluate robustness of our automated pipeline to different hyperparameters, we further analyzed the automatic labels using k=100 clusters (see new appendix F.2). We found that the most distinguishing clusters between Phi-4 and Phi-4-Reasoning (with the largest Chi square difference) reflect the differences in algorithmic approach we had previously identified (Table 6). In particular, the top cluster in Phi-4-Reasoning expressed a nearest-neighbor approach, whereas the top cluster in Phi-4 expressed a brute-force approach. Crucially, we did not have to manually analyze these clusters at all and were simply able to draw these conclusions from the automated labelling pipeline. Overall, this shows that 1) our hand-identified algorithmic primitives are robust to both automation and (reasonable) choices of hyperparameters, and 2) our novel labeling pipeline enables automated end-to-end clustering and primitive identification.
> >
> > We thank the reviewer for highlighting this issue, which allowed us to significantly advance our automated end-to-end pipeline. We hope that the novel additions adequately address their concern.

---

> > > ### Author Response · Authors · 2025-11-21
> > >
> > > > some clusters (e.g., compare_or_verify) appear context-dependent and multi-purpose.
> > >
> > > The entanglement and context-dependency of multiple primitives could have a number of causes. First, entanglement may reflect how certain primitives often cooccur or are chained together in reasoning. For instance, Compare_or_verify primitives often occur after path_generation, so the entanglement of their representations may mirror their functional relationship. Thus, the entanglement may follow reasoning motif structure, where primitives involved in a recurring multi-step motif (sequence of multiple primitives) are likely to be more entangled. This suggests the context-dependency may capture meaningful algorithmic patterns, not just noise. Second, some primitives may have computational similarity. For instance, given the hierarchical nature of primitives, they may share a number of computational sub-primitives. Third, a given primitive such as Compare_or_verify evokes different sub-routines or sub-primitives depending on the context, e.g., it compares distances after generating paths or verifies the accuracy of distance computation. While the second point mostly concerns cross-primitive similarity as a source of entanglement, the third point regards context-dependency of the same primitive.
> > >
> > > We thank the reviewer for this helpful question, and will add our explanation to the discussion section of the paper after the discussion period.
> > >
> > > > Can we have time complexity for these methods?
> > >
> > > Our time complexity is determined by two main processes:
> > > (i) CoT Generation + Activation Collection: $O(L^2)$, where L = sequence length (number of tokens in the reasoning trace). Complexity is quadratic because Transformer self-attention computes all token-to-token interactions. This happens once per response to generate the trace and extract internal activations. Typical responses contain between 500 and 6000 tokens; at that scale, the activation collection is the most expensive step for our method, taking between 5-10 minutes on 4 A40 GPUs (we note that multiple GPUs are required to analyze long responses).
> > >
> > > (ii) Clustering: O(K × N × I), with K = number of clusters (K=50 usually), N = number of datapoints (activations collected), I = iterations until convergence, Linear scaling with each factor. This is much faster than activation collection: we measured 61 seconds for 10 responses with 50 clusters (linear, ~1 min with a CPU).
> > >
> > > We have added an Appendix B.7 to explain this.
> > >
> > > We hope that our responses and revisions effectively addressed your concerns raised and thank you again for your helpful review. We would be happy to provide any additional clarifications and continue the discussion.

---

### Author Response · Authors · 2025-12-03
**Summary of the rebuttal phase**

Dear Area Chair,

We would like to summarize the key interactions with the reviewers and how we addressed their comments during the rebuttal period. We thank the reviewers once again for their positive evaluation of our paper and constructive feedback, which helped us to further strengthen the paper by automating our approach and clarifying our formal and methodological choices and contributions in more detail.

**Summary of our additional analyses and our responses**

Below we summarize our major improvements during the rebuttal:

1. Reviewers asked us for further clarification about our choices of hyperparameters. In response, we conducted a number of thorough ablations to evaluate the robustness of our method to a) the choice of layer, b) the choice of the number of clusters K, and c) the number of responses (see Detailed Responses, section 1 below).


2. Reviewers appreciated our findings and approach, but asked whether the functional role of the different clusters in latent space could merely be manually interpreted. In response, we have provided an automated LLM-based pipeline for labeling these clusters and compared the outcome with our previous methods. The findings align well with the manually assigned labels (see Detailed Responses, section 2 below). Two reviewers engaged with our rebuttal, one increased their response from 6 to an 8, and the other from a 2 to a 6.

3. Reviewer NeZR asked about a control in order to rule out a bag-of-words steering explanation of our primitive injection. That is, they asked whether our injected primitive vectors largely increase the likelihood of a bag of tokens rather than a latent procedure. We have conducted and added this control and show that it cannot account for the observed effects (see Detailed Responses, section 3 below). In response, this reviewer raised their rating from a 2 to a 6.

**Summary of reviewer feedback**

In the initial reviews, reviewers highlighted a number of positive aspects about our paper:

They found the discovery of compositional algorithmic primitives compelling:
 - Tj9J: “Additive composition… suggesting shareable, reusable building blocks.”
 - NEzR: “Treats skills as additive residual directions… compositional tests.”

They emphasized that the paper addresses a meaningful and relevant problem:
 - Tj9J: “The problem this paper is exploring is interesting to me.”
 - EaF2: “This work suggests that general algorithmic building blocks are used by LLMs… the topic is hence relevant to the community.”
 - 7Ty5: “Particularly interesting and offers new insights into LLM reasoning mechanisms.”

They viewed the framework as principled, grounded, and extensible:
 - Tj9J: “Delivers a genuinely end-to-end interpretability account.”
 - EaF2: “Straightforward, driven by first principles… application to other settings seems straightforward.”
 - 7Ty5: “A novel and thought-provoking framework.”

They also pointed to strong empirical support for the approach:
 - Tj9J: “Vector injection and subtraction produce clear, selective changes in reasoning hallmarks.”
 - 7Ty5: “Experiments cover multiple models and datasets.”


**Results of discussion period**

Before the discussion period, three reviewers (Tj9J, EaF2, 7Ty5) were already in favor of acceptance (each assigning a score of 6). All three raised specific concerns that we directly addressed in our rebuttal analyses (see our individual responses and our overview above).
Reviewer NEzR initially assigned a 2 but stated that several clarifications and additions “could” raise their score “to a 4 or better.” After reviewing our new analyses, NEzR reported that their “main concerns … are now well addressed” and that they would increase their score to 6.

Reviewer 7Ty5 similarly confirmed that their concern “has been addressed” and stated their intention to increase their score. Reviewers Tj9J and EaF2 did not have the opportunity to respond due to the shortened discussion period, but their reviews voiced closely related concerns regarding the choice of hyperparameters (see point 1 above) and manual labeling (see point 2). Both reviewers were originally in favor of accepting the paper (score 6).

---

> ### Author Response · Authors · 2025-12-03
>
> **Detailed responses**
>
> Please note that we also respond to all reviewer questions in detail in response to each review. Below we provide a brief overview of key additions we made during the rebuttal.
>
> **1 Hyperparameter choices**
>
> All reviewers asked us to better justify our choices of hyperparameters, asking about the choice of layer, the choice of the number of clusters, and the number of samples used to train the clustering algorithm. In response, we repeated the clustering analysis for different layers, numbers of clusters, and numbers of responses.  Below, we provide a brief summary of our findings (for further details, see the responses to the reviewers and Appendix E).
>
> 1. To analyze robustness to our choice of layers, we repeated the clustering analysis for all layers in both Phi4 and Phi4-R for TSP (Traveling Salesperson Problem). We found two groups of correlated layers, broadly layers 10-25 and layers 30-40. In particular, the earlier group was more focused on reasoning and associated hierarchy (which we were interested in characterizing in our investigation). This justified layer 17 as a suitable candidate for clustering across models and responses (see Appendix E.1).
> 2. To evaluate robustness to our choice of K, we conducted a similar analysis, which revealed K=50 as the parsimonious and functionally appropriate choice, offering a reasonable tradeoff between explaining variance in the latent space and parsimony in the potential number of algorithmic primitives (see Appendix E.2).
> 3. To evaluate robustness to the number of responses, we repeated the clustering analysis using different numbers of responses, revealing high consistency for 5, 25, and 50 responses (see Appendix E.3).
>
> **2 Hand-interpreted mapping**
>
> Reviewers highlighted that we manually interpreted the function of different clusters and manually selected representative clusters to further analyze, and asked about further details about our approach. This allowed us to significantly improve our paper by offering an automated pipeline that yields similar results to our manual approach. We have now included a novel automated pipeline for labeling primitive clusters using LLMs (description in Appendix F) and its results. This allows for (i) the automated labeling of identified clusters, and (ii) the verification of manually assigned labels. We found that the automatically extracted labels agree with the manually assigned roles (see Table below, annotations for all fifty clusters are added to the Appendix, Table 5).
>
> | Manual Label            | Automated (Token-centric)                               | Automated (Context-centric)                                           |
> |-------------------------|----------------------------------------------------------|------------------------------------------------------------------------|
> | `produce_final_answer`  | Output formatting with \<final_answer\> tag               | Output formatting with \<final_answer\> tag.                             |
> | `compare_or_verify_2`   | Route exploration and modification suggestions           | Route evaluation and improvement through swaps and recalculations.     |
> | `nearest_neighbor`      | "Selecting best path based on edge weights"              | "Selecting best path based on edge costs."                             |
> | `compare_or_verify`     | Proposed alternative routes                              | Route optimization and evaluation in TSP solutions.                    |
> | `compute_distance`      | Summation and accumulation of path costs                 | Addition and summation of route distances.                             |
> | `generate_new_path`     | Proposed routes for alternative solutions                | Route exploration and evaluation for optimization.                     |
>
>
> Crucially, this pipeline can be integrated in an automated end-to-end implementation of our framework, using a novel metric we propose to quantify the expressivity of each cluster (see Appendix A.7). Notably, we find that our manual labels are distinguished by very high expressivity (see Table 5). To evaluate robustness of our automated pipeline to different hyperparameters, we further analyzed the automatic labels using K=100 clusters, finding that the most distinguishing clusters between Phi-4 and Phi-4-Reasoning had reflected the differences in algorithmic approach we had previously identified (Table 6 and Appendix F.2). Overall, this shows that 1) our hand-identified algorithmic primitives are robust to both automation and (reasonable) choices of hyperparameters, and 2) our novel labeling pipeline enables automated end-to-end clustering and primitive identification.

---

> > ### Author Response · Authors · 2025-12-03
> >
> > **3 Bag-of-words explanation**
> >
> > Reviewer NEzR highlighted the following concern:
> >
> >
> > > Alternative explanation (bag-of-words steering). Line 218 suggests K-Means is run over (h_17) for tokens in self-generated reasoning traces. If so, clusters may mostly track lexical/discourse cues (“alternatively”, “thus”, parentheses/digits) rather than procedure. The resulting average vector could simply boost a bag of tokens, which in turn nudges the next steps of a “reasoning” template. This also explains linear add/subtract as logit mass addition/suppression—not necessarily algorithmic composition. Suggest to add controlled experiments (e.g., unembedding alignment) to show this reflects a latent procedure rather than BOW steering.
> >
> > To address this, we included a new control where we inject the primitive vector at the end of the residual stream, directly before the unembedding. This control allowed us to test whether the effects we observed could simply be explained by shifting the logits toward a set of tokens associated with a primitive vector, rather than by a latent procedure. When controlling for this (by subtracting the maximal effect obtained by injecting the primitive vectors at the end of the residual stream), we still observe the same effects, confirming that they are caused by a latent procedure (see Table 3 and Appendix D.1 in the paper).

---

### Meta-Review · Area_Chair_HLxa · 2025-12-27

**Summary:**

This submission proposes a mechanistic framework to identify “algorithmic primitives” in LLM reasoning by clustering internal activations and constructing “primitive vectors” that can be injected into the residual stream to induce/suppress reasoning steps and to test linear compositionality across tasks and models.

Reviewer sentiment is overall positive but initially borderline: three reviewers (Tj9J, EaF2, 7Ty5) rated the paper 6 (“marginally above the acceptance threshold”), with concerns around (i) sensitivity to clustering hyperparameters / layer choice and sampling, (ii) manual interpretation/selection of clusters and questions about disentanglement/causal meaning, and (iii) presentation/reproducibility gaps (notably a missing clear definition of how primitive vectors are computed and incomplete figure labeling).

During rebuttal, the authors added new analyses: broad ablations over layer choice / K / number of responses and quantitative stability metrics; an automated LLM-based labeling pipeline that matches the manual labels; additional controls to rule out a bag-of-words/logit-bias explanation; and time complexity estimates.

I still lean reject: the key “algorithmic primitives” claim is not yet convincingly established beyond a clustering-and-injection pipeline that appears sensitive to methodological choices and interpretive labeling, and remaining clarity/reproducibility gaps make independent verification difficult at this stage.

**Reviewer Concerns:**

Concerns addressed:

- Hyperparameter sensitivity / robustness (layer choice, K, sampling noise): originally flagged by multiple reviewers. Authors added extensive layer/K/response-count ablations + stability metrics (incl. ARI/NMI-style clustering consistency).

- Manual labeling / cluster selection validity: reviewers asked whether the cluster-to-primitive mapping is hand-interpreted and potentially context-dependent. Authors added an automated LLM-based labeling pipeline and show agreement with manual labels, plus a proposed “cluster expressivity” metric to guide automated selection.

- Bag-of-words / “logit mass shift” alternative explanation: raised strongly by NEzR. Authors added a control injecting at the end of residual stream before unembedding and report the effect persists after controlling for that baseline, supporting a latent-procedure interpretation.

- Time complexity: requested explicitly.  Authors added a clear breakdown (generation/activation collection vs clustering) with measured runtimes and an appendix pointer.


Concerns could still be improved

- Presentation / readability polish: at least one reviewer emphasized presentation gaps (axis labels/captions; self-contained definitions) as a blocker for assessment/reproducibility.

- one reviewer questioned whether primitives are sufficiently independent and asked for a more comprehensive analysis of dependencies/causal relations.

**Reviewer Scores:**

I cannot reliably answer this counterfactual question without putting words in reviewers’ mouths. I will not impute score changes beyond what reviewers explicitly stated in the discussion. I instead provide a faithful synthesis of the discussion outcomes and remaining points of disagreement.

---

### Decision · Program_Chairs · 2026-01-26

Reject